# Attack by Yourself: Effective and Unnoticeable Multi-Category Graph Backdoor Attacks with Subgraph Triggers Pool

**Jiangtong Li**[*,1,2], **Dongyi Liu**[*,1], **Kun Zhu**[1,2], **Dawei Cheng**[#,1,2], **Changjun Jiang**[#,1,2],

1. Key Laboratory of Embedded System and Service Computing,
Ministry of Education, Tongji University
2. School of Computer Science and Technology,
Tongji University
{jiangtongli, 2152833, kzhu00, dcheng, cjjiang}@tongji.edu.cn

## Abstract

**G**raph **N**eural **N**etworks (GNNs) have achieved significant success in various real-world applications, including social networks, finance systems, and traffic management. Recent researches highlight their vulnerability to backdoor attacks in node classification, where GNNs trained on a poisoned graph misclassify a test node only when specific triggers are attached. These studies typically focus on single attack categories and use adaptive trigger generators to create node-specific triggers. However, adaptive trigger generators typically have a simple structure, limited parameters, and lack category-aware graph knowledge, which makes them struggle to handle backdoor attacks across multiple categories as the number of target categories increases. We address this gap by proposing a novel approach for **E**ffective and **U**nnoticeable **M**ulti-**C**ategory (EUMC) graph backdoor attacks, leveraging subgraph from the attacked graph as category-aware triggers to precisely control the target category. To ensure the effectiveness of our method, we construct a **M**ulti-**C**ategory **S**ubgraph **T**riggers **P**ool (MC-STP) using the subgraphs of the attacked graph as triggers. We then exploit the attachment probability shifts of each subgraph trigger as category-aware priors for target category determination. Moreover, we develop a "select then attach" strategy that connects suitable category-aware trigger to attacked nodes for unnoticeability. Extensive experiments across different real-world datasets confirm the efficacy of our method in conducting multi-category graph backdoor attacks on various GNN models and defense strategies. Code is released at `https://github.com/novdream/EUMC`.

## 1 Introduction

In recent years, GNNs [1, 2] have achieved significant success in modeling real-world graph-structured data, including social networks [3], financial interactions [4, 5], and traffic flows [6]. GNNs typically update node representations by aggregating information from their neighbors, which preserves the features of the neighbors and captures the local graph topology. However, GNNs are vulnerable to attacks, and numerous methods have been developed to deceive these networks [7].

In graph adversarial attacks, adversaries can modify existing nodes or edges in the graph (**G**raph **M**odification **A**ttack, GMA) [8] or inject malicious nodes to the graph (**G**raph **I**njection **A**ttack, GIA) [9] in evasion or poison settings. For instance, TDGIA [10] traverses the original graph to

---

[*]Both authors contributed equally to this research. [#]Corresponding Authors

39th Conference on Neural Information Processing Systems (NeurIPS 2025).

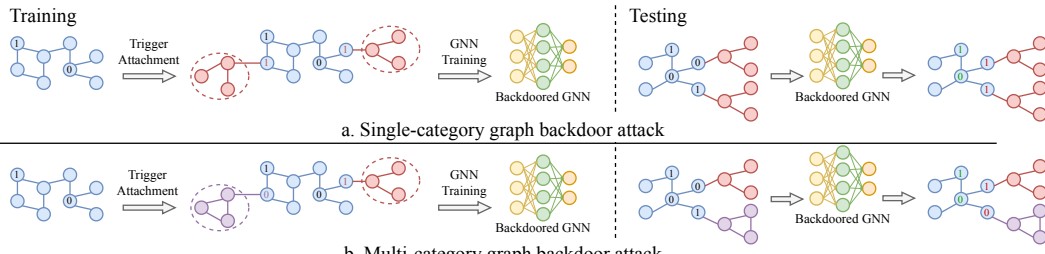

Figure 1: The comparison between single-category and multi-category graph backdoor attack on node classification.

Table 1: The ASR(%) of UGBA against defenses on Flickr as the number of target categories changes.

| Defense | 1 | 2 | 3 | 4 | 5 | 6 | 7 |
|---------|------|------|------|------|------|------|------|
| None | 99.7 | 98.1 | 65.5 | 49.2 | 34.6 | 29.9 | 24.7 |
| Prune | 99.4 | 99.0 | 66.0 | 49.5 | 37.1 | 26.7 | 24.4 |
| Prune+LD | 99.3 | 98.5 | 71.8 | 50.3 | 38.0 | 30.1 | 23.7 |

identify vulnerable nodes using a topological defective edge strategy and generates features for injected nodes with a smooth feature optimization objective. However, these attacks often lead to suboptimal efficacy and challenges in achieving specific objectives [10]. Furthermore, they usually change the homogeneity of the attacked graph and require large attack budgets [11], making the alterations easily detectable. The visibility not only diminishes the efficiency needed for effective adversarial strategies but also limits their practical applicability unnoticeable scenarios.

To address these issues, developing graphs backdoor attacks is a promising approach, which typically unfold in three steps. Initially, adversaries create a poisoned graph by attaching trigger to a small set of nodes known as poisoned samples and assigning them the label of the target category. Subsequently, when GNNs are trained on this poisoned graph, they learn to associate the trigger with the target category. In the inference, only the nodes that are linked with the triggers are predicted as the corresponding target category, while clean nodes are predicted as usual. Compared to graph adversarial attacks, graph backdoor attacks offer three primary advantages: 1) *Lower Computational Cost*: the attack requires no additional optimization during inference; 2) *More Precise Target Control*: the triggers directly influence the prediction of the target category; 3) *Higher Unnoticeability*: the backdoor activates only when specific triggers appear.

Recent works have primarily focused on enhancing backdoor attacks for graph classification, yet node classification remains underexplored. GTA [12] explores an adaptive trigger generator to create more powerful, node-specific triggers for node classification. To reduce attack budgets and enhance the unnoticeability of backdoor attacks, UGBA [13] selects representative nodes as poison nodes and optimizes the adaptive trigger generator with an additional unnoticeable loss and DPGBA [14] exploit the GAN loss to preserve the feature distribution within the generated triggers. These graph backdoor attack methods focus on targeting a specific category, allowing backdoored models to consistently produce a predetermined malicious category when triggers are attached. However, they lack the capability to effectively attack across multiple categories, *i.e.*, manipulating the model to predict different target categories for the same node using various triggers. This limitation primarily stems from the use of adaptive trigger generators designed for backdoor attacks. These generators usually have a simple structure and limited trainable parameters, which constrain their ability to generate triggers for multiple attacked target categories. Furthermore, these generators lack category-aware graph knowledge, such as understanding how different subgraphs influence classification results on various nodes. As a result, they struggle to optimize multiple adaptive trigger generators for different target categories as the number of target categories increases. In Table 1, we further empirically confirm this limitation on Flickr [15].

In this paper, we address the complex issue of multi-category graph backdoor attacks on node classification, as depicted in Figure 1. We introduce the **E**ffective and **U**nnoticeable **M**ulti-**C**ategory (EUMC) graph backdoor attack method, which utilizes influential subgraphs from the attacked graph as triggers. Our approach constructs a **M**ulti-**C**ategory **S**ubgraph **T**riggers **P**ool (MC-STP) to manage different target categories, effectively circumventing the optimization challenges associated with multiple trigger generators. Specifically, we start by sampling hundreds of subgraphs from the attacked graph

to establish the base of our MC-STP. For each subgraph, we calculate the attachment probability shifts by assessing whether the subgraph is attached to given nodes. This process identifies the influential subgraphs and determines the target categories for each, as depicted in Figure 2 and discussed in Section 4.1. To ensure unnoticeability, we further develop a "select then attach" strategy that selects the suitable subgraph trigger from the pool and properly connects it to each attacked node. Note that the subgraph triggers for backdoor attack are from the original graph, therefore, the feature distribution can also be well preserved. Through these steps, EUMC achieves control over node classifications within the graph backdoor attack, manipulating them effectively. Extensive experiments across six real-world datasets confirm the efficacy of our method in conducting multi-category graph backdoor attacks on various GNNs and defense strategies. Our main contributions can be summarized as

- We study a novel and challenging problem of effective and unnoticeable multi-category graph backdoor attack.
- We design a framework that constructs a MC-STP from the attacked graph and develop a "select then attach" strategy to ensure the unnoticeability and effectiveness of the multi-category graph backdoor attacks.
- Extensive experiments on node classification datasets demonstrate the efficacy of our method in multi-category graph backdoor attacks on various GNN models and defense strategies.

## 2 Related Works

### 2.1 GNNs on Node Classification

GNNs are highly effective in node classification tasks, playing a crucial role in interpreting graph-structured data. These networks leverage relational information between nodes through a message-passing mechanism, updating node features based on their neighbors' information [16]. This process captures both local node features [17] and global structural context [18, 19], enabling accurate node classification in complex networks [20]. GNNs encode node and topological features through iterative aggregation and feature transformation, enhancing the precision of node classification. Recent advancements include the development of sophisticated aggregation functions [21, 22] that capture nuanced node interactions. Attention mechanisms [23] and transformer layers [24] refine the aggregation process, improving model adaptability. Moreover, multi-scale feature extraction methods [25] allow GNNs to consider various neighborhood sizes, enriching representational capability and boosting classification performance across diverse datasets. In this paper, we utilize GCN [16], GAT [23], and GraphSAGE [26] to exemplify node classification networks in our backdoor attack studies.

### 2.2 Adversarial Attacks on GNN

Based on the type of manipulation on the graph, graph adversarial attacks can be categorized into GMA [27] and GIA [28]. In GMA, attackers deliberately alter existing nodes or edges within the graph to compromise the model's integrity. These modifications aim to degrade the performance of GNNs by inducing misclassifications or erroneous predictions. Conversely, GIA involves introducing fake nodes and edges into the graph. This method is particularly effective as it allows attackers to add new structural data specifically designed to mislead the GNN without altering the original graph's properties. Common optimization techniques for these adversarial attacks on graphs include the use of gradient descent [27, 10] to find optimal perturbations and reinforcement learning [8, 28] to dynamically adjust attack strategies based on the GNNs' responses.

### 2.3 Backdoor Attacks on GNN

Backdoor attacks on GNNs typically insert triggers into the training graph and assign the desired target label to samples containing these triggers. Consequently, a model trained on this poisoned graph is deceived when it processes test samples with specific triggers. Backdoor attacks vary by tasks and learning paradigm, like graph classification [29], node classification [12], graph contrastive learning [30], and graph prompt learning [31]. For graph classification, a common method transforms edges and nodes into a predefined subgraph [32], while Xu and Picek [33] enhance unnoticeability by assigning triggers without altering the labels of poisoned samples. In node classification, efforts to manipulate node and edge features for backdoor attacks often face practical challenges, as changing

existing nodes' links and attributes is typically outside control of the attackers [34]. Furthermore, GTA [12] employs an adaptive trigger generator to create specific triggers, UGBA [13] uses representative nodes to refine this generator for unnoticeability, and DPGBA [14] exploit the GAN loss to preserve the feature distribution within the generated triggers. However, adaptive trigger generators often have a simple structure and limited parameters, lacking category-aware graph knowledge to manage multiple category backdoor attacks as the number of target categories grows. Our work develops an effective and unnoticeable multi-category graph backdoor attack, enabling the attacker to control the predicted categories with varied triggers. In detail, we first constructs a MC-STP, using attachment probability shifts as category priors, and then develops a "select then attach" strategy that connects suitable trigger to attacked nodes, ensuring the unnoticeability and effectiveness.

## 3 Preliminary

### 3.1 Node Classification

We represent a graph with $\mathcal{G} = (\mathcal{V}, \mathbf{A}, \mathbf{X})$, where $\mathcal{V} = \{v_1, ..., v_N\}$ denotes the set of nodes, $\mathbf{A} \in \mathbb{R}^{N \times N}$ is the adjacency matrix of graph $\mathcal{G}$, and $\mathbf{X} = \{\mathbf{x}_1, ..., \mathbf{x}_N\}$ represents the features of nodes with $\mathbf{x}_i$ corresponding to $v_i$. Here, $\mathbf{A}_{ij} = 1$ indicates a connection between nodes $v_i$ and $v_j$; otherwise, $\mathbf{A}_{ij} = 0$. The node classification task takes graph $\mathcal{G}$ as input and outputs labels for each node in $\mathcal{V}$. Formally, the GNN node classifier $f_\theta : \mathcal{G}^i \rightarrow \{1, 2, ..., K\}$, where $\mathcal{G}^i$ represents the computation graph for node $v_i$ and $\{1, 2, ..., K\}$ is the set of possible labels. The classifier $f_\theta$ generates node feature iteratively by aggregating the feature of its neighbors, with the final neural network layer outputting a label for different nodes. In this paper, we focus on the semi-supervised node classification task in inductive setting. The whole graph is split into labeled graph $\mathcal{G}_L = (\mathcal{V}_L, \mathbf{A}_L, \mathbf{X}_L)$ and unlabeled graph $\mathcal{G}_U = (\mathcal{V}_U, \mathbf{A}_U, \mathbf{X}_U)$ with no overlap, $\mathcal{V}_L \cap \mathcal{V}_U = \emptyset$.

### 3.2 Threat Model

**Attacker's Goal** The goal of the attacker is to mislead the GNN to classify target nodes with attached trigger as the target category. Moreover, for the same target node, the attacker can manipulate the GNN model to predict different target categories by attaching category-aware trigger. At the same time, the GNN should function normally for clean nodes that do not have triggers attached.

**Attacker's Knowledge and Capability** In most backdoor attack scenarios, the training data for the target model is accessible to attackers, although the architecture of the target GNN model remains unknown to them. Attackers can attach trigger and label to nodes within a budget before training the target models to poison the training graph. In inference, attackers can attach trigger to the target node.

### 3.3 Backdoor Attacks for Node Classification

The fundamental concept of backdoor attacks involves linking a trigger with the target category in the training data, causing target models to misclassify during inference. As depicted in Figure 1, in training, an attacker attaches a trigger $t$ to a subset of nodes $\mathcal{V}_P \subset \mathcal{V}_U$ and assigns them the target category label $y_t$. More details about the $\mathcal{V}_P$ and attachment strategy of our method are in Section 4.

GNNs trained on this backdoored dataset learn to associate the presence of trigger $t$ with the target category $y_t$. During testing, attaching trigger $t$ to a test node $v$ causes the backdoored GNN to classify $v$ as category $y_t$. To evaluate the performance of our backdoor attack, the labeled graph $\mathcal{G}_L$ is split into training graph $\mathcal{G}_{Tr}$, validation graph $\mathcal{G}_{Va}$, and test graph $\mathcal{G}_{Te}$ with no overlap Earlier efforts [12, 13] have advanced backdoor attacks on node classification by creating node-specific triggers or adjusting node and edge features for smoothness [35].

Our work addresses the challenging problem of multi-category graph backdoor attacks, as illustrated in Figure 1 (b). **For each node in poisoned nodes set $\mathcal{V}_P$, we attach it with a category-aware trigger $t_k$ and assign corresponding target categories $y_k$.** Consequently, during the test phase, different triggers $t_k$ can misclassify a test node $v$ into the corresponding target category $y_k$. Unlike methods focused on a single target category, our approach creates diverse triggers tailored to multiple target categories. Existing methods employ an adaptive trigger generator to generate the trigger for a single target category. In Table 1, we extend UGBA [13] by increasing the number of target categories from 1 to 7 on the Flickr dataset with multiple adaptive trigger generators to handle different target

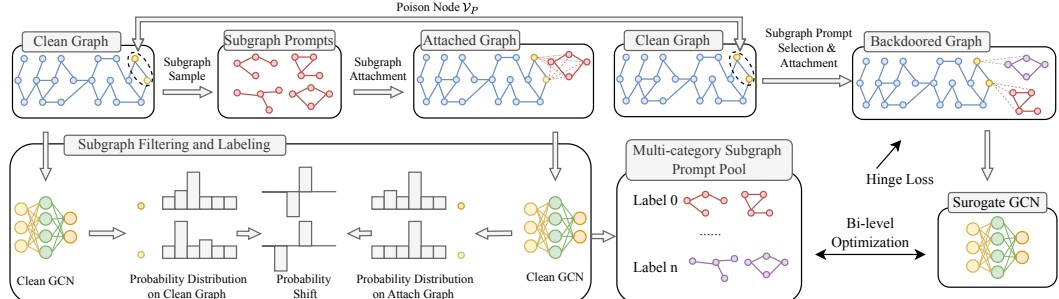

Figure 2: The illustration of our method on multi-category graph backdoor attack.

categories. Experimental results demonstrate that effectiveness diminishes as the number of target categories increases. To address these challenges, we employ category-aware subgraphs as triggers to enhance the effectiveness of multi-category graph backdoor attacks. Formally, the multi-category graph backdoor attack for node classification is defined as follows:

**Definition 1** *Given a clean training graph $\mathcal{G}_{Tr} = (\mathcal{V}_{Tr}, \mathbf{A}_{Tr}, \mathbf{X}_{Tr})$ and corresponding labels $\mathcal{Y}_{Tr}$, and another subset $\mathcal{V}_U$ without labels, our goal is to optimize a MC-STP, $\mathbf{P} = \{y_1 : [t_1^1, ..., t_1^{n_{pool}}]; ...; y_K : [t_K^1, ..., t_K^{n_{pool}}]\}$. We explore a trigger attacher $a_\phi$ to select trigger $t_k^i$ from $\mathbf{P}_{y_k}$ and attach them to poisoned nodes $\mathcal{V}_P$. The training objective is that a GNN $f_\theta$ trained on the poisoned graph will classify a test node attached with trigger $t_k^i$ into target category $y_k$:*

$$\min_{\mathbf{P}} \sum_{v_i \in \mathcal{V}_U} \sum_{k \in \{1,...,K\}} l(f_{\theta^*}(a_\phi(\mathcal{G}_P^i, \mathbf{P}_{y_k})), y_k)$$

$$s.t. \theta^* = \arg\min_\theta \sum_{v_i \in \mathcal{V}_{Tr}} l(f_\theta(\mathcal{G}_{Tr}^i, y_i)) + \sum_{v_i \in \mathcal{V}_P} \sum_{k \in \{1,...,K\}} l(f_\theta(a_\phi(\mathcal{G}_P^i, \mathbf{P}_{y_k})), y_k),$$

*where $l(\cdot)$ is the cross entropy loss for node classification. The architecture of the target GNN $f$ is unknown and may include various defense mechanisms.*

## 4  Methodology

In this section, we detail our method, designed to optimize Eq. 1 for unnoticeable and effective multi-category graph backdoor attacks, as depicted in Figure 2. Our approach involves a MC-STP $\mathbf{P}$, a trigger attacher $a_\phi$, and a surrogate GCN model $f_s$. Initially, we sample several subgraphs from the clean training graph $\mathcal{G}_{Tr}$. Each of these subgraphs is then inserted into pre-selected representative nodes of the clean training graph, and a GCN, trained on this graph, calculates the probability shift before and after the insertion. Based on this shift, we select influential subgraphs and determine the target label for each to initialize the MC-STP. The trigger attacher then selects an unnoticeable subgraph trigger for each poisoned node $\mathcal{V}_P$ and attaches this trigger effectively to deceive $f_s$. To ensure the effectiveness of the multi-category graph backdoor attack, we employ a bi-level optimization [12, 13] with the surrogate GCN model.

### 4.1  Multi-category Subgraph Triggers Pool

As discussed in Section 3.3, adaptive trigger generators [13] struggle as the number of target categories increases. Therefore, we construct a MC-STP to effectively execute backdoor attacks on multi-category graph backdoor attacks. To this end, we initially randomly select several nodes from the unlabeled graph, $\mathcal{V}_U$, to serve as central nodes. We then employ the **B**readth-**F**irst **S**earch (BFS) algorithm to sample subgraphs around each central node, which form the basis of our MC-STP. To ensure that the sampled subgraphs are influential for node classification and provide sufficient misleading priors for backdoor attack, we develop an algorithm based on attachment probability shifts. This algorithm filters the subgraphs, assigning each influential subgraph with a target category.

Specifically, we first train a clean two-layer GCN network, $f_{\theta_c}$, on the training graph $\mathcal{G}_{Tr}$ for node classification on $n_y$ categories, where $n_y$ is number of target categories. This GCN is utilized to assess

the misleading effects of different subgraph triggers. Subsequently, we apply K-Means clustering, as described in UGBA [13], to select representative nodes $\mathcal{V}_P$ as poisoned nodes, where the sizes of $\mathcal{V}_P$ for different datasets are shown in Table 2. These poisoned nodes $\mathcal{V}_P$ are then used to calculate the attachment probability shifts. During this process, we attach each candidate subgraph $t$ to all nodes in $\mathcal{V}_P$ and use $f_{\theta_c}$ to compute the variance in prediction probability for $\mathcal{V}_P$ before and after the attack as the **A**ttachment **P**robability **S**hift (APS), which is defined as:

$$\text{APS}_t = \frac{1}{|\mathcal{V}_P|} \sum_{v_i \in \mathcal{V}_P} f_{\theta_c}(a_\phi(\mathcal{G}_P^i, t)) - f_{\theta_c}(\mathcal{G}_P^i) \tag{1}$$

where $a_\phi(\cdot)$ is the trigger attacher, $\mathcal{G}_P^i$ is the computation graph for node $v_i$, and $\text{APS}_t \in \mathcal{R}^{n_y}$ is a vector to measure the probability shift among different target categories.

The APS quantifies the misleading effect of each subgraph trigger. Therefore, we select subgraphs with $\max(\text{APS}_t) > 0.2$ as influential triggers, and the attack target category is then determined by $c_t = \arg\max(\text{APS}_t)$. Then we retain the top-$n_{pool}$ subgraph triggers (measured by $\max(APS_t)$) in each target category to form the MC-STP, $\mathbf{P} = \{y_1 : [t_1^1, ..., t_1^{n_{pool}}]; ...; y_K : [t_K^1, ..., t_K^{n_{pool}}]\}$. In this way, the MC-STP introduces category-aware prior, indicating the graph can attack itself. On the one hand, it contains a rich set of subgraph triggers for various attack scenarios; on the other hand, each target category is controlled by influential subgraphs, facilitating the optimization.

## 4.2 Trigger Selection and Attachment

To effectively inject the trigger from the MC-STP, $\mathbf{P}$, we address two critical questions: 1) which trigger from the pool should be used for backdoor attack given the target category; and 2) how the trigger should be attached to attacked nodes. To ensure the triggers remain unnoticeable, we develop a "select then attach" strategy based on node similarity. For unnoticeable and effective backdoor attack, we first select the subgraph trigger by comparing cosine similarity between the subgraph features and the attacked node features. For a poison node $v_i \in \mathcal{V}_P$, the selection is formulated by:

$$s_{t_{y_k}^j} = \sum_{\mathbf{x}_k \in \bar{\mathbf{X}}_{t_{y_k}^j}} \frac{\mathbf{x}_k \cdot \mathbf{x}_i}{||\mathbf{x}_k||_2 ||\mathbf{x}_i||_2}, \tag{2}$$

where $\mathbf{x}_i$ is the feature of $v_i$. The trigger with the highest $s_{t_{y_k}^j}$ is selected for backdoor attack.

Our method focuses on conducting unnoticeable and effective graph backdoor attacks. To ensure a basic backdoor attack, it is necessary to attach at least one node from the subgraph trigger to the attacked node. To enhance the unnoticeability of the graph backdoor attack, we need to attach nodes from the subgraph trigger that exhibit relatively high similarity to the attacked node. To improve the effectiveness of the graph backdoor attack, more nodes from the subgraph trigger are attached to the attacked node. Therefore, the trigger attacher, $a_\phi(\cdot)$, operates by: 1) calculating the similarity between the subgraph trigger and the attacked node; 2) attaching the node with the highest similarity from the subgraph trigger to the attacked node; 3) attaching nodes with a similarity greater than $\tau_a$ from the subgraph trigger to the attacked node, where $\tau_a$ is the similarity threshold. This approach balances the effectiveness and unnoticeability of the backdoor attack when attaching subgraph trigger.

## 4.3 Optimization

To ensure the effectiveness and unnoticeability of the graph backdoor attack, we optimize the MC-STP, $\mathbf{P}$ through a bi-level optimization to successfully attack the surrogate GCN model $f_s$. The training of the surrogate GCN $f_s$ on the poisoned graph is formulated as:

$$\min_{\theta_s} \mathcal{L}_s(\theta_s, \theta_P) = \sum_{v_i \in \mathcal{V}_{Tr}} l(f_s(\mathcal{G}_{Tr}^i, y_i)) + \sum_{v_i \in \mathcal{V}_P} \sum_{k \in \{1,...,K\}} l(f_s(a(\mathcal{G}_P^i, \mathbf{P}_{y_k})), y_k), \tag{3}$$

where $\theta_s$ and $\theta_P$ are the parameters of the surrogate GCN and MC-STP. $\mathcal{G}_{Tr}^i$ is the clean labeled training graph for node $v_i$ with label $y_i$, and $y_k$ is the attack target label of $a(\mathcal{G}_P^i, \mathbf{P}_{y_k})$. For effective misleading, $\mathbf{P}$ and $a_\phi$ must induce surrogate model to classify nodes with triggers as target category:

$$\mathcal{L}_a = \sum_{v_i \in \mathcal{V}_U} \sum_{k \in \{1,...,K\}} l(f_s(a(\mathcal{G}_P^i, \mathbf{P}_{y_k})), y_k), \tag{4}$$

**Algorithm 1** Algorithm of EUMC.

---

**Input**: Original graph $\mathcal{G}$, target category set $\mathcal{Y}$=1, ..., K
**Parameter**: $\alpha$
**Output**: MC-STP (**P**), Backdoored Graph ($\mathcal{G}_B$)

1: Initialize $\mathcal{G}_B = \mathcal{G}$;
2: Separate the training graph $\mathcal{G}_{Tr}$ from labeled graph $\mathcal{G}_L$;
3: Select poisoned nodes $\mathcal{V}_P$ based on the cluster algorithm from UGBA [13];
4: Randomly initialize $\theta_s$ for $f_s$;
5: Initialize **P** with $\theta_P$ as parameter based on the construction of MC-STP in Section 4.1;
6: **while** not converged **do**
7:     **for** t=1,2,...,N **do**
8:         Update $\theta_s$ by $\nabla_{\theta_s} \mathcal{L}_s$ based on Eq.3;
9:     **end for**
10:     Update $\theta_P$ by $\nabla_{\theta_P}(\mathcal{L}_a + \alpha\mathcal{L}_h)$ based on Eq.6;
11: **end while**
12: **for** $v_i \in \mathcal{V}_p$ **do**
13:     Random select the attack target category $y_k$ from $\mathcal{Y}$;
14:     Update $\mathcal{G}_B$ with $a_\phi(G_B^i, P_{y_k},)$ based on the "select then attach" strategy in Section 4.2;
15: **end for**
16: **return P** and $\mathcal{G}_B$;

---

Moreover, all attached nodes should closely resemble the attacked node, formulated as:

$$\mathcal{L}_h = \sum_{v_i \in \mathcal{V}_P} \sum_{(v_i, v_j) \in \mathcal{E}_t^i} \max(0, \tau_L - \cos(v_i, v_j)), \tag{5}$$

where $\mathcal{E}_t^i$ comprises all edges connecting the attached nodes from the subgraph trigger to node $v_i$, $\tau_L$ is the similarity threshold, and $\cos(\cdot)$ is the cosine similarity. Thus, we formulate the following bi-level optimization problem with balance hyper-parameter $\alpha$:

$$\min_{\theta_P} \mathcal{L}_p(\theta_s^*, \theta_P) = \mathcal{L}_a + \alpha\mathcal{L}_h \qquad \text{s.t.} \quad \theta_s^* = \arg\min_{\theta_s} \mathcal{L}_s(\theta_s, \theta_P). \tag{6}$$

To reduce computation cost, $\mathcal{L}_a$ in Eq. 4 is calculated by randomly assigning a target category $y_k$ to $v_i$. Therefore, the computational complexity is irrelative to the number of target category. **More details about the analysis of time complexity can be found in supplementary materials.** We adopt the bi-level optimization [13] to update Eqs. 3 and 6 as outlined in Algorithm 1.

## 5   Experiments

### 5.1   Experimental Settings

**Datasets.** To demonstrate the effectiveness of our method, we conduct experiments on six public real-world datasets: Cora, Pubmed [36], Bitcoin [37], Facebook [38], Flickr [15], and OGB-arxiv [39]. These datasets are widely used for inductive semi-supervised node classification. Cora and Pubmed are small-scale citation networks, while Bitcoin represents an anonymized network of Bitcoin legality transactions. Facebook is characterized by page-page relationships, Flickr links image captions that share common properties, and OGB-arxiv is a large-scale citation network. The statistics of these datasets are provided in Table 3a.

**Compared Methods.** We compare our method with representative and state-of-the-art graph backdoor attack methods, including SBA [29], GTA [12], UGBA [13], and DPGBA [14]. We apply Prune and Prune+LD [13] for attribute-based defense, which prune edges based on node similarity. We also apply OD [14] for distribution-based defense, which trains a outlier detector (*i.e.*, DOMINANT [40]) to filter out outlier nodes. Hyper-parameters are determined based on the validation performance. **More details are in supplementary material.**

**Evaluation Protocol.** Following a similar setup as in UGBA, we randomly mask 20% of the nodes from the original dataset. Half of these masked nodes are designated as target nodes for evaluating

Table 2: Backdoor attack results (ASR (%) | CA (%)). Only CA is reported for clean graphs.

| Dataset | $|\mathcal{V}_P|$ | Defense | Clean | SBA | GTA | UGBA | DPGBA | EUMC |
|---|---|---|---|---|---|---|---|---|
| Cora | 100 | None | 82.5 | 14.3 \| 82.9 | 87.7 \| 77.4 | 83.1 \| 73.4 | 87.3 \| 82.5 | **97.4** \| 82.4 |
| | | Prune | 81.4 | 14.3 \| 81.8 | 15.2 \| 80.2 | 84.8 \| 72.9 | 84.3 \| 82.3 | **93.5** \| 82.2 |
| | | Prune+LD | 80.0 | 14.3 \| 79.5 | 15.1 \| 79.5 | 85.3 \| 72.1 | 80.1 \| 81.2 | **93.5** \| 81.0 |
| | | OD | 83.2 | 14.3 \| 83.5 | 65.0 \| 81.9 | 86.6 \| 79.8 | 87.9 \| 83.1 | **96.8** \| 81.9 |
| Pubmed | 150 | None | 84.3 | 33.3 \| 85.1 | 86.9 \| 84.8 | 88.9 \| 84.7 | 91.5 \| 85.3 | **96.4** \| 83.9 |
| | | Prune | 83.7 | 33.3 \| 85.3 | 33.4 \| 84.9 | 92.6 \| 84.5 | 89.5 \| 85.0 | **95.0** \| 83.9 |
| | | Prune+LD | 84.2 | 33.3 \| 83.6 | 33.4 \| 83.7 | 85.8 \| 83.7 | 92.1 \| 84.5 | **95.3** \| 83.4 |
| | | OD | 84.9 | 33.3 \| 85.3 | 88.6 \| 85.4 | 89.7 \| 85.4 | 89.5 \| 85.1 | **96.2** \| 84.6 |
| Bitcoin | 300 | None | 78.3 | 33.3 \| 78.3 | 79.2 \| 78.3 | 76.4 \| 78.3 | 80.1 \| 78.3 | **90.6** \| 78.3 |
| | | Prune | 78.3 | 33.3 \| 78.3 | 37.6 \| 78.3 | 73.3 \| 78.3 | 33.3 \| 78.3 | **88.3** \| 78.3 |
| | | Prune+LD | 78.3 | 33.3 \| 78.3 | 42.6 \| 78.3 | 72.0 \| 78.3 | 33.3 \| 78.3 | **86.5** \| 78.3 |
| | | OD | 78.3 | 33.3 \| 78.3 | 54.1 \| 78.3 | 56.1 \| 78.3 | 77.6 \| 78.3 | **88.4** \| 78.3 |
| Facebook | 100 | None | 82.3 | 25.0 \| 86.6 | 76.0 \| 85.9 | 84.9 \| 85.8 | 86.6 \| 85.8 | **91.7** \| 83.8 |
| | | Prune | 81.5 | 25.0 \| 86.2 | 25.5 \| 85.6 | 86.4 \| 85.5 | 72.4 \| 85.3 | **91.7** \| 83.3 |
| | | Prune+LD | 81.0 | 25.0 \| 86.0 | 25.3 \| 85.3 | 73.9 \| 84.6 | 75.4 \| 85.2 | **92.0** \| 83.3 |
| | | OD | 83.3 | 25.0 \| 86.0 | 77.2 \| 85.5 | 82.4 \| 85.9 | 85.4 \| 85.9 | **92.5** \| 84.3 |
| Flickr | 300 | None | 45.9 | 14.3 \| 46.3 | 93.1 \| 42.6 | 24.6 \| 43.4 | 53.5 \| 45.2 | **90.4** \| 44.5 |
| | | Prune | 45.1 | 14.3 \| 42.8 | 15.0 \| 41.5 | 24.4 \| 42.9 | 68.6 \| 44.8 | **90.3** \| 44.1 |
| | | Prune+LD | 44.5 | 14.3 \| 45.0 | 14.6 \| 44.2 | 23.7 \| 42.6 | 18.2 \| 45.0 | **90.8** \| 44.2 |
| | | OD | 46.1 | 14.3 \| 43.2 | 26.4 \| 41.9 | 18.1 \| 44.1 | 79.7 \| 44.4 | **89.6** \| 44.6 |
| OGB-arxiv | 800 | None | 64.8 | 2.5 \| 66.2 | 68.4 \| 65.6 | 63.7 \| 64.9 | 68.8 \| 64.9 | **83.8** \| 65.3 |
| | | Prune | 64.9 | 2.5 \| 64.6 | 3.1 \| 65.0 | 69.3 \| 65.1 | 13.0 \| 65.0 | **84.1** \| 65.5 |
| | | Prune+LD | 63.9 | 2.5 \| 64.9 | 3.3 \| 65.3 | 69.4 \| 65.2 | 6.2 \| 65.0 | **84.2** \| 65.3 |
| | | OD | 64.9 | 2.5 \| 64.5 | 28.2 \| 63.8 | 62.6 \| 65.2 | 54.1 \| 65.0 | **83.4** \| 65.1 |

attack performance, while the other half serve as clean test nodes to assess the prediction accuracy of backdoored models on normal samples. The graph containing the remaining 80% of nodes is used as the training graph $\mathcal{G}_{Tr}$, with the 20% nodes $\mathcal{V}_L$ are labeled. We use the average **A**ttack **S**uccess **R**ate (ASR) on the target node set across different target categories and clean accuracy on clean test nodes to evaluate the effectiveness of the backdoor attacks. To demonstrate the transferability of the backdoor attacks, we target GNNs with varying architectures, namely **GCN**, **GraphSage**, and **GAT**. We conduct experiments on each target GNN architecture five times and report the average performance. **Further details about the time complexity analysis are in supplementary materials.**

**Implementation Details.** A 2-layer GCN is deployed as the surrogate model for all datasets. All hyper-parameter are determined based on the performance on the validation set. Specifically, $\alpha$, trigger pool size, the trigger size, hidden dimension and inner iterations step is set as 5, 40, 5, 64 and 5, respectively. Moreover, $\tau_L$ is set as 0.4, 0.4, 1.0, 0.5, 0.6, and 0.8 for Cora, Pubmed, Bitcoin, Fackbook, Flickr and OGB-arxiv, respectively. $\tau_a$ is set as 0.2, 0.2, 0.8, 0.2, 0.2, and 0.8 for Cora, Pubmed, Bitcoin, Fackbook, Flickr and OGB-arxiv, respectively. The pruning threshold Prune and Prune+LD defense is set to filter out 10% most dissimilar edges Therefore, the thresholds are 0.1, 0.1, 0.8, 0.2, 0.2, 0.8 for Cora, Pubmed, Bitcoin, Fackbook, Flickr and OGB-arxiv, respectively.

## 5.2 Main Results

In Table 2, we compare with other four graph backdoor method on six real-world datasets to validate the effectiveness and unnoticeability of our method under different defense strategy, *i.e.*, None, Prune, and Prune + LD. The ASR and Clean Accuracy are averaged on three GNN architectures: GCN, GAT, and GraphSAGE. **Detailed comparisons for each GNN model are in supplementary material.**

In terms of clean accuracy, EUMC achieves results comparable to other baseline models and GNNs trained on clean graphs, indicating that graph backdoor attacks minimally impact clean nodes due to the small proportion of poisoned nodes. From the perspective of ASR, EUMC excels on the Cora, Flickr, and OGB-arxiv datasets, achieving state-of-the-art performance. The large number of target categories (> 5) in these datasets demonstrates the effectiveness of EUMC in multi-category graph backdoor attacks. Moreover, EUMC outperforms other models on Bitcoin, Facebook, and Pubmed datasets, further validating the effectiveness of EUMC across varying target category numbers.

Comparing different defense strategies, EUMC demonstrates high effectiveness and balanced performance, indicating its ability to evade defenses and launch unnoticeable attacks. Comparing EUMC with UGBA and DPGBA on Flickr and OGB-arxiv, EUMC performs less effectively on the category-rich OGB-arxiv, which is consistent with expectations for multi-category attacks, *i.e.*, the larger the

Table 3: (a) Dataset Statistics; (b) Ablation Study

| Dataset | #Nodes | #Edge | #Feature | #Classes |
|---|---|---|---|---|
| Cora | 2,708 | 5,429 | 1443 | 7 |
| Pubmed | 19,717 | 44,338 | 500 | 3 |
| Bitcoin | 203,769 | 234,355 | 165 | 3 |
| Facebook | 22,470 | 342,004 | 128 | 4 |
| Flickr | 89,250 | 899,756 | 500 | 7 |
| OGB-arxiv | 169,343 | 1,166,243 | 128 | 40 |

| Setting | None | Prune | Prune+LD |
|---|---|---|---|
| w/o stru | $82.0 \pm 17.3$ | $70.8 \pm 29.6$ | $78.4 \pm 23.3$ |
| w/o feat | $61.5 \pm 37.3$ | $66.1 \pm 33.8$ | $43.4 \pm 34.6$ |
| w/o tgt | $67.4 \pm 34.9$ | $66.6 \pm 36.0$ | $79.3 \pm 21.7$ |
| w/o sele | $69.8 \pm 35.6$ | $74.0 \pm 22.3$ | $77.7 \pm 24.2$ |
| link all | $79.7 \pm 14.1$ | $75.8 \pm 18.2$ | $88.8 \pm 12.9$ |
| link one | $75.3 \pm 25.2$ | $76.1 \pm 23.2$ | $75.9 \pm 23.7$ |
| full | $89.4 \pm 9.7$ | $88.8 \pm 10.3$ | $89.9 \pm 9.1$ |

(a) Statistics on the Cora, Pubmed, Bitcoin, Facebook, Flickr, and OGB-arxiv datasets.

(b) The average $\pm$ standard deviation ASR(%) of EUMC under alternative setting on the Flickr dataset.

number of target categories, the lower the achievable ASR. Conversely, UGBA and DPGBA exhibits weaker performance on Flickr, highlighting the vulnerability of adaptive trigger generator-based methods under unbalanced conditions. Overall, EUMC effectively and unnoticeably attack GNNs in multi-category setting, outperforming existing methods across diverse defense and datasets.

## 5.3 Ablation Study

To valid the effectiveness of the subgraph triggers pool and "select then attach" strategy, we conduct experiment on the Flickr dataset and report the average and standard deviation ASR(%) of GCN, GAT, and GraphSAGE. **More ablation studies can be found in the supplementary material.**

**Construction of MC-STP.** To validate the effectiveness of our MC-STP, we conduct experiments with different construction settings, *i.e.*, randomly initializing the subgraph structure (w/o stru), the subgraph features (w/o feat), and the target category (w/o target) as shown in Table 3b. The results indicate that each of these settings significantly contributes to attack effectiveness, demonstrating the robustness of our MC-STP. Notably, node features from the clean graph have the most significant impact among them, underscoring the importance of node features in classification.

**"Select and Attach" Strategy.** To further validate the effectiveness of our "select then attach" strategy, we conduct experiments with three alternatives: randomly selecting the subgraph trigger from the pool (w/o sele), attaching all nodes to the attacked node (link all), and attaching the most similar node to the attacked node (link one), as shown in Table 3b. The results demonstrate that our "select then attach" strategy outperforms all others. Both attaching more nodes or fewer nodes affects performance under different defense strategies, confirming the efficacy of our approach.

## 5.4 Similarity Analysis

We explore the impact of different attack types on edge similarity in poisoned graphs. Specifically, we assess the effects of GTA, UGBA, EUMC, and EUMC without $\mathcal{L}_h$ on edge similarity between trigger edges (connected to trigger nodes) and clean edges (not connected to trigger nodes) on the OGB-arxiv dataset, as shown in Figure 3. The results reveal that in EUMC without $\mathcal{L}_h$ setting, the similarity between trigger and clean edges still exceeds that of GTA, due to our "select then attach" strategy. With the application of $\mathcal{L}_h$, this similarity further increases to the level of UGBA. This demonstrates that the unnoticeability of EUMC arises from both $\mathcal{L}_h$ and "select then attach" strategy.

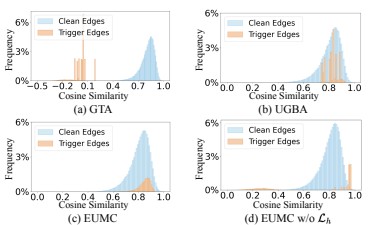

Figure 3: Edge similarity distributions on OGB-arxiv.

## 6 Statistical Significant Test

We perform the significance test between our EUMC and the strongest baseline, *i.e.* DPGBA [14]. We run our EUMC and DPGBA $5\times4\times3=60$ times with three GNNs (GCN, GAT, and GraphSage) and four defense strategy with random seeds ranging from 1 to 5. The ASR of our method are $95.3\pm2.1$ and $83.8\pm10.6$ in Core and OGB-arixv dataset, while the ASR of DPGBA are $84.9\pm6.7$ and $56.8\pm30.7$ in Core and OGB-arixv dataset. At the significance level 0.05, we perform significance test to verify that our method is better than DPGBA. The p-values in Core and OGB-arixv dataset are

$6.32 \times 10^{-21}$ and $2.70 \times 10^{-9}$, which is far below 0.05, which demonstrates that the superiority of our method is statistically significant.

# 7    Limitation and Future Direction

While our method achieves strong attack success across multiple classification scenarios, we observe a slight decrease in performance as the number of target classes grows. Although this degradation remains marginal and does not compromise the overall effectiveness, future work could explore scalable trigger generation strategies or class-agnostic optimization techniques to ensure stable performance regardless of label granularity. Moreover, our current focus has been predominantly on the attack side of backdoor research, with comparatively limited attention to defensive mechanisms. In follow-up studies, we plan to investigate lightweight yet robust countermeasures, such as real-time anomaly detection, adaptive sanitization pipelines, or defense-aware trigger designs, to complement our attack framework and advance a more holistic understanding of backdoor vulnerabilities.

# 8    Broader Impacts

By advancing the study of backdoor attacks in graph neural networks, our work lays the groundwork for more realistic evaluations of model vulnerabilities in practical deployments. Demonstrating robust multi-category insertion techniques not only underscores potential risks in real-world applications but also serves as a importunate for the defensive research community to evolve beyond single-class mitigation schemes. In particular, our findings encourage the design of next-generation defenses that can detect and neutralize backdoors across arbitrarily large label spaces, ultimately contributing to safer and more trustworthy AI systems.

# 9    Conclusion

In this paper, we tackle the complex challenge of conducting effective and unnoticeable multi-category graph backdoor attacks on node classification. We show that existing backdoor attacks, which rely on adaptive trigger generators, are not effective for managing multi-category attacks as the number of target categories increases. To address this, we construct a multi-category subgraph triggers pool from the subgraphs of the attacked graph and utilized attachment probability shifts as category-aware priors for subgraph trigger selection and target category determination. Moreover, we develop a "select then attach" strategy that connects appropriate trigger to attacked nodes, ensuring unnoticeability. Extensive experiments on real-world graph node classification datasets show the effectiveness and unnoticeability of our method in controlling various GNNs for multi-category graph backdoor attacks.

## Acknowledgments and Disclosure of Funding

This work is supported in part by the National Key Research and Development Program of China (Grant No. 2022YFB4501704), in part by the National Natural Science Foundation of China (Grant No. 62402341, 62472317, 62302337), in part by the Shanghai "Scientific and Technological Innovation Action Plan" High and New Technology Projects (22YS1400600), in part by the Postdoctoral Fellowship Program of CPSF (GZC20241225), and in part by the China Postdoctoral Science Foundation (2025M771513).

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

## A    Details of Compared Methods

The details of the compared methods are described as:

- **SBA [29]**: This method targets backdoor attacks on graph classification by injecting a fixed subgraph as a trigger into the training graph for a poisoned node. The edges of each subgraph are generated using the Erdos-Renyi (ER) model, and the node features are randomly sampled from the training graph, ensuring variability in the features of the injected subgraph.

- **GTA [12]**: This method addresses backdoor attacks on both graph and node classification. It starts by randomly selecting unlabeled nodes from the clean graph as poison nodes. An adaptive trigger generator is then used to create node-specific subgraphs as triggers. The trigger generator is optimized through a bi-optimization algorithm that incorporates backdoor attack loss.

- **UGBA [13]**: Similar to GTA, UGBA focuses on backdoor attacks on node classification and employs an adaptive trigger generator to generate node-specific triggers. To enhance the unnoticeability of the attack, UGBA introduces a clustering algorithm to select representative nodes as poison nodes. This method also explores the use of an unnoticeable loss function to increase the similarity between attacked nodes and generated triggers, improving the stealthiness of the backdoor attacks.

- **DPGBA [14]**: DPGBA focuses on in-domain (ID) trigger generation for the backdoor attacks on node classification. To generate ID triggers, DPGBA introduce an out-of-distribution (OOD) detector in conjunction with an adversarial learning strategy to generate the attributes of the triggers within distribution. This method further introduces novel modules designed to enhance trigger memorization by the victim model trained on poisoned graph.

## B    Details of Defense Strategies

The details of defense strategies are described as follows:

- **Prune**: In this strategy, we focus on enhancing the resilience of GNNs to graph backdoor attacks by pruning edges that connect nodes with low cosine similarity. This approach is based on the observation that edges created by backdoor attackers often link nodes with dissimilar features, aiming to manipulate the model's predictions subtly. By pruning such edges, we can potentially disrupt the structure of the trigger inserted by the attacker, making it less effective and thus preserving the integrity of the data representation of graph.

- **Prune+LD**: Building upon the Prune strategy, this approach adds an extra defense against backdoor attacks by addressing the issue of "dirty" label on nodes that may have been compromised by the attacker. In addition to pruning edges between dissimilar nodes, we also discard the labels of these nodes to mitigate the influence of potentially poisoned labels. This dual approach helps in further safeguarding the learning process against manipulation. By removing these labels, the defense mechanism reduces the risk of the model learning from and perpetuating the attacker's modifications, thereby maintaining the performance and trustworthiness of GNNs in the face of adversarial conditions.

Table 4: Training Time on OGB-arxiv

| Metrics | GTA | UGBA | DPGBA | EUMC |
|---|---|---|---|---|
| ASR(None) | 68.4 | 63.7 | 68.8 | 83.8 |
| ASR(Prune) | 3.1 | 69.3 | 13.0 | 84.1 |
| ASR(Prune+LD) | 3.3 | 69.4 | 6.2 | 84.2 |
| ASR(OD) | 28.2 | 62.6 | 54.1 | 83.4 |
| Time | 86.4s | 98.5s | 121.4s | 117.2s |

Table 5: Results of backdooring GCN (ASR (%) | Clean Accuracy (%)). Only clean accuracy is reported for clean graph.

| Datasets | $\mathcal{V}_P$ | Defense | Clean | SBA | GTA | UGBA | DPGBA | EUMC |
|---|---|---|---|---|---|---|---|---|
| Cora | 100 | None | 82.5 | 14.3±0.0 \| 83.7±1.2 | 93.4±3.4 \| 71.0±7.0 | 91.2±3.9 \| 63.3±2.1 | 93.2±0.3 \| 82.1±0.9 | **96.9±0.4** \| 82.0±0.2 |
| | | Prune | 80.6 | 14.3±0.0 \| 82.5±1.3 | 15.6±1.8 \| 80.6±0.8 | 88.8±3.9 \| 64.4±3.9 | 89.3±0.2 \| 81.1±0.6 | **93.2±0.2** \| 81.9±0.3 |
| | | Prune+LD | 80.4 | 14.3±0.0 \| 80.4±1.2 | 15.0±0.7 \| 80.1±1.5 | 88.8±1.8 \| 69.0±2.9 | 87.8±1.1 \| 82.7±0.6 | **93.0±0.4** \| 81.4±0.6 |
| | | OD | 82.1 | 14.3±0.0 \| 83.0±1.3 | 69.0±24.1 \| 82.0±1.1 | 88.7±2.3 \| 79.1±2.6 | 93.4±0.4 \| 83.6±0.2 | **95.4±0.3** \| 82.4±0.4 |
| Pubmed | 150 | None | 83.7 | 33.3±0.0 \| 85.3±0.2 | 90.8±2.2 \| 85.2±0.1 | 96.3±0.7 \| 84.7±0.2 | 95.3±0.7 \| 84.8±0.1 | **96.6±1.1** \| 83.7±0.3 |
| | | Prune | 83.1 | 33.3±0.0 \| 85.5±0.2 | 33.4±0.1 \| 85.1±0.2 | **97.9±0.5** \| 84.7±0.2 | 93.2±0.1 \| 84.5±0.1 | 95.3±0.2 \| 83.6±0.1 |
| | | Prune+LD | 84.2 | 33.3±0.0 \| 84.1±0.3 | 33.5±0.1 \| 84.0±0.1 | 95.5±0.4 \| 83.8±0.1 | 94.2±0.3 \| 84.2±0.2 | **95.6±0.3** \| 83.3±0.3 |
| | | OD | 84.7 | 33.3±0.0 \| 85.3±0.4 | 93.4±0.8 \| 85.9±0.2 | 95.5±0.2 \| 84.9±0.1 | 93.5±0.4 \| 84.6±0.1 | **97.1±0.1** \| 84.2±0.2 |
| Bitcoin | 300 | None | 78.2 | 33.3±0.0 \| 78.3±0.0 | 84.0±14.1 \| 78.3±0.0 | 91.8±10.8 \| 78.3±0.0 | 87.6±9.9 \| 78.3±0.0 | **96.4±3.3** \| 78.3±0.0 |
| | | Prune | 78.2 | 33.3±0.0 \| 78.3±0.0 | 38.3±6.1 \| 78.3±0.0 | 85.2±9.1 \| 78.3±0.0 | 33.3±0.1 \| 78.3±0.0 | **92.9±7.1** \| 78.3±0.0 |
| | | Prune+LD | 78.2 | 33.3±0.0 \| 78.3±0.0 | 45.0±11.8 \| 78.3±0.0 | 79.6±9.3 \| 78.3±0.0 | 33.3±0.1 \| 78.3±0.0 | **97.3±1.7** \| 78.3±0.0 |
| | | OD | 78.3 | 33.3±0.0 \| 78.3±0.0 | 51.7±12.8 \| 78.3±0.0 | 66.6±8.3 \| 78.3±0.0 | 86.2±0.6 \| 78.3±0.0 | **97.6±1.2** \| 78.3±0.0 |
| Facebook | 100 | None | 81.3 | 25.0±0.0 \| 85.9±0.3 | 81.9±5.8 \| 84.8±0.4 | 93.8±0.3 \| 84.5±0.2 | 95.1±0.8 \| 84.1±0.2 | **95.9±0.3** \| 81.9±0.5 |
| | | Prune | 79.9 | 25.0±0.0 \| 85.9±0.2 | 25.4±0.8 \| 84.5±0.4 | 94.2±0.5 \| 84.4±0.4 | 76.5±0.2 \| 83.6±0.4 | **96.3±0.1** \| 82.6±0.4 |
| | | Prune+LD | 79.7 | 25.0±0.0 \| 85.4±0.3 | 25.3±0.4 \| 84.1±0.6 | 91.8±0.4 \| 82.9±0.3 | 80.1±0.2 \| 83.4±0.3 | **96.2±0.4** \| 81.8±0.6 |
| | | OD | 84.1 | 25.0±0.0 \| 85.2±0.2 | 86.2±3.3 \| 84.4±0.4 | 93.1±0.5 \| 84.7±0.2 | 93.1±0.2 \| 84.4±0.1 | **94.5±0.3** \| 82.9±0.4 |
| Flickr | 300 | None | 46.1 | 14.3±0.0 \| 45.6±0.2 | **99.8±0.1** \| 41.9±0.8 | 21.4±0.6 \| 41.0±0.4 | 56.5±0.3 \| 44.0±0.1 | 98.6±0.2 \| 43.8±0.3 |
| | | Prune | 45.5 | 14.3±0.0 \| 42.0±0.8 | 15.5±0.7 \| 40.5±0.1 | 18.6±0.7 \| 41.1±0.5 | 71.5±0.6 \| 43.5±0.2 | **99.5±0.2** \| 43.9±0.2 |
| | | Prune+LD | 46.1 | 14.3±0.0 \| 45.2±0.3 | 14.9±0.5 \| 43.8±0.4 | 21.9±0.8 \| 40.4±0.1 | 18.4±1.6 \| 44.3±0.2 | **98.5±0.7** \| 43.7±0.3 |
| | | OD | 46.2 | 14.3±0.0 \| 42.3±0.3 | 28.6±0.1 \| 41.0±0.1 | 17.8±0.2 \| 42.3±0.2 | 85.7±0.8 \| 43.6±0.1 | **99.4±0.5** \| 43.1±0.2 |
| OGB-arixv | 800 | None | 64.1 | 2.5±0.0 \| 66.1±0.4 | 68.4±1.9 \| 65.2±0.3 | 74.2±1.2 \| 64.5±0.6 | 73.7±0.9 \| 64.8±0.1 | **78.3±0.6** \| 65.3±0.3 |
| | | Prune | 64.2 | 2.5±0.0 \| 64.5±0.2 | 3.4±0.4 \| 64.5±0.4 | 70.8±0.9 \| 64.5±0.6 | 16.9±0.5 \| 64.9±0.1 | **78.6±0.6** \| 65.4±0.4 |
| | | Prune+LD | 64.0 | 2.5±0.0 \| 65.2±0.2 | 3.3±0.6 \| 64.7±0.3 | 71.0±1.1 \| 64.9±0.5 | 10.2±1.1 \| 64.8±0.1 | **79.0±0.9** \| 65.9±0.2 |
| | | OD | 64.5 | 2.5±0.0 \| 64.5±0.1 | 35.7±1.7 \| 63.3±0.2 | 66.3±1.5 \| 64.9±0.2 | 74.1±0.7 \| 64.6±0.2 | **77.9±0.5** \| 65.0±0.4 |

- **OD**: In this strategy, we focus on enhancing the resilience of GNNs to graph backdoor attacks by removing OOD nodes in the graph. This approach is based on the observation that the features from triggers often have different node feature distribution from the clean data. Therefore, the OOD detector (*i.e.*, DOMINANT [40]) trained on the poisoned graph can identify the trigger by the reconstruction loss. By removing the nodes with high reconstruction losses, the feature distribution of the nodes can be distributed in domain, preserving the integrity of the data representation of graph.

## C  Time Complexity Analysis

During the bi-level optimization phase, the computation cost of each outer iteration consist of updating of surrogate GCN model in inner iterations and optimizing multi-category subgraph trigger pool. Let $h$ denote the embedding dimension. The cost for updating the surrogate model is $O(Nhd|\mathcal{V}|)$, where $d$ is the average degree of nodes, $N$ is the number of inner iterations for the surrogate model, and $|\mathcal{V}|$ is the size of training nodes and poisoned nodes. For the optimization of multi-category subgraph trigger pool, the cost for optimizing $\mathcal{L}_\alpha$ is $O(hd(|\mathcal{V}_U|))$, where $|\mathcal{V}_U|$ is the size of unlabeled nodes. And the cost for optimizing $\mathcal{L}_h$ is $O(h(|\mathcal{V}_P| * |\mathcal{V}_a|))$, where $|\mathcal{V}_P|$ is the number of poisoned nodes and $|\mathcal{V}_a|$ is the number of attached nodes. Note that, during optimizing $\mathcal{L}_\alpha$, we randomly assigning a target category $y_k$ to poisoned nodes, making the time complex reduced from $O(Khd|\mathcal{V}_P|)$ to $O(hd|\mathcal{V}_P|)$, where $K$ is the number of target categories. Since $|\mathcal{V}_P| \ll |\mathcal{V}|$, $|\mathcal{V}_P| * |\mathcal{V}_a| \ll |\mathcal{V}|$ and $|\mathcal{V}_U| \approx |\mathcal{V}|$, the overall time complexity for each outer iteration is $O((N+1)hd|\mathcal{V}|)$, which is similar to time complextity of UGBA [13]. In the backdoor attack phase, the cost of selecting and attaching trigger to the target node is $O(hn_{pool})$. Our time complexity analysis proves that EUMC has great potential in large-scale applications.

In Table 4, we also report the overall training time of EUMC, DPGBA, UGBA and GTA on OGB-arixv dataset. All models are trained with 200 epochs on an A100 GPU with 80G memory. Experimental

Table 6: Results of backdooring GAT (ASR (%) | Clean Accuracy (%)). Only clean accuracy is reported for clean graph.

| Datasets | $\mathcal{V}_P$ | Defense | Clean | SBA | | GTA | | UGBA | | DPGBA | | EUMC | |
|---|---|---|---|---|---|---|---|---|---|---|---|---|---|
| Cora | 100 | None | 84.6 | 14.3±0.0 | 84.6±1.4 | 70.7±5.1 | 81.2±1.1 | 66.1±9.9 | 77.9±2.1 | 81.8±7.0 | 81.6±2.7 | **97.7±1.0** | 82.4±0.2 |
| | | Prune | 83.9 | 14.3±0.0 | 83.9±1.3 | 14.8±0.7 | 81.7±1.0 | 74.4±6.1 | 78.7±1.3 | 80.1±8.4 | 82.1±1.3 | **94.2±0.7** | 81.9±0.1 |
| | | Prune+LD | 81.3 | 14.3±0.0 | 81.5±1.3 | 15.0±0.9 | 80.6±1.8 | 79.6±8.7 | 75.9±2.7 | 72.2±3.9 | 80.9±0.2 | **94.0±0.6** | 81.2±0.3 |
| | | OD | 83.7 | 14.3±0.0 | 85.8±1.1 | 61.0±14.6 | 82.8±2.8 | 78.9±3.4 | 79.8±2.9 | 83.8±2.7 | 82.3±0.2 | **97.2±1.6** | 82.0±0.8 |
| Pubmed | 150 | None | 85.1 | 33.3±0.0 | 84.0±0.3 | 87.1±3.0 | 83.7±0.2 | 91.4±0.6 | 83.9±0.2 | 90.7±0.9 | 84.5±0.1 | **96.6±1.0** | 83.0±0.2 |
| | | Prune | 83.1 | 33.3±0.0 | 83.7±0.4 | 33.4±0.1 | 83.4±0.3 | **94.8±0.7** | 83.5±0.1 | 90.7±0.6 | 84.2±0.1 | 94.7±2.4 | 82.8±0.4 |
| | | Prune+LD | 83.3 | 33.3±0.0 | 82.9±0.5 | 33.4±0.1 | 83.4±0.4 | 91.3±0.3 | 83.2±0.2 | 93.4±0.7 | 83.7±0.2 | **95.8±0.8** | 82.2±0.5 |
| | | OD | 84.9 | 33.3±0.0 | 84.4±0.1 | 87.8±1.6 | 84.3±0.2 | 90.1±0.2 | 84.8±0.2 | 89.1±0.6 | 84.4±0.2 | **96.1±0.1** | 83.9±0.2 |
| Bitcoin | 300 | None | 78.3 | 33.3±0.0 | 78.3±0.0 | 69.4±10.4 | 78.2±0.1 | 49.3±15.1 | 78.3±0.0 | 72.0±2.1 | 78.3±0.0 | **76.2±8.6** | 78.3±0.0 |
| | | Prune | 78.2 | 33.3±0.0 | 78.3±0.0 | 39.3±7.4 | 78.3±0.0 | 53.2±18.0 | 78.3±0.0 | 33.3±0.0 | 78.3±0.0 | **72.9±9.3** | 78.3±0.0 |
| | | Prune+LD | 78.2 | 33.3±0.0 | 78.3±0.0 | 42.8±12.7 | 78.3±0.0 | 55.9±13.9 | 78.3±0.0 | 33.3±0.0 | 78.3±0.0 | **63.2±2.4** | 78.3±0.0 |
| | | OD | 78.3 | 33.3±0.0 | 78.3±0.0 | 66.0±0.4 | 78.3±0.0 | 68.3±38.1 | 78.3±0.0 | 58.7±0.2 | 78.3±0.0 | **68.4±0.5** | 78.3±0.0 |
| Facebook | 100 | None | 79.5 | 25.0±0.0 | 87.1±0.3 | 64.2±2.0 | 86.3±0.4 | 75.5±3.1 | 86.2±0.3 | 79.6±4.8 | 86.1±0.2 | **92.3±3.0** | 83.4±0.2 |
| | | Prune | 78.7 | 25.0±0.0 | 86.0±0.2 | 25.7±0.7 | 85.8±0.3 | 79.3±3.8 | 85.7±0.2 | 70.7±1.9 | 85.5±0.3 | **91.9±2.8** | 81.2±0.1 |
| | | Prune+LD | 78.3 | 25.0±0.0 | 85.9±0.2 | 25.3±0.4 | 85.7±0.2 | 52.1±0.9 | 85.1±0.1 | 72.8±2.4 | 85.2±0.2 | **92.5±2.8** | 81.9±0.1 |
| | | OD | 80.1 | 25.0±0.0 | 86.0±0.1 | 69.1±6.6 | 86.1±0.4 | 81.8±0.4 | 86.2±0.3 | 78.5±4.3 | 86.2±0.5 | **95.9±0.7** | 83.6±0.4 |
| Flickr | 300 | None | 46.7 | 14.3±0.0 | 45.6±0.4 | **79.9±19.5** | 40.5±0.3 | 23.7±0.9 | 43.1±1.4 | 49.7±5.9 | 44.6±0.4 | 78.1±9.1 | 44.7±0.3 |
| | | Prune | 44.9 | 14.3±0.0 | 40.4±0.0 | 14.5±0.5 | 40.5±0.3 | 19.9±2.3 | 42.5±0.4 | 67.6±3.1 | 44.3±0.2 | **76.5±9.0** | 43.9±0.4 |
| | | Prune+LD | 42.4 | 14.3±0.0 | 45.4±0.8 | 14.2±0.4 | 45.1±0.4 | 21.1±1.8 | 42.1±0.4 | 15.3±2.6 | 44.8±0.1 | **78.3±6.6** | 43.2±0.3 |
| | | OD | 46.4 | 14.3±0.0 | 40.2±0.0 | 25.7±5.2 | 40.2±0.1 | 17.6±0.4 | 43.4±0.1 | 68.4±6.0 | 43.5±0.2 | **75.2±6.1** | 44.4±0.9 |
| OGB-arxiv | 800 | None | 65.6 | 2.5±0.0 | 66.1±0.1 | 68.4±1.9 | 65.5±0.2 | 73.6±1.7 | 65.4±0.2 | 59.4±9.8 | 64.6±0.1 | **98.6±0.1** | 64.8±0.1 |
| | | Prune | 65.6 | 2.5±0.0 | 64.1±0.3 | 2.9±0.5 | 65.9±0.1 | 82.7±0.5 | 65.5±0.1 | 9.0±0.6 | 64.7±0.2 | **98.8±0.1** | 65.1±0.2 |
| | | Prune+LD | 64.4 | 2.5±0.0 | 64.9±0.3 | 3.4±1.0 | 66.4±0.2 | 83.8±1.2 | 65.6±0.2 | 2.8±0.1 | 65.2±0.2 | **99.2±0.2** | 64.3±0.3 |
| | | OD | 65.4 | 2.5±0.0 | 64.2±0.1 | 24.2±2.2 | 64.5±0.3 | 83.5±0.5 | 65.0±0.4 | 13.2±7.0 | 64.6±0.1 | **98.3±0.7** | 64.4±0.4 |

Table 7: Results of backdooring GraphSAGE (ASR (%) | Clean Accuracy (%)). Only clean accuracy is reported for clean graph.

| Datasets | $\mathcal{V}_P$ | Defense | Clean | SBA | | GTA | | UGBA | | DPGBA | | EUMC | |
|---|---|---|---|---|---|---|---|---|---|---|---|---|---|
| Cora | 100 | None | 80.5 | 14.3±0.0 | 80.4±1.3 | **99.0±0.8** | 80.1±0.8 | 92.1±1.5 | 79.1±1.7 | 86.9±1.3 | 83.7±0.1 | 97.5±1.8 | 82.8±0.2 |
| | | Prune | 79.6 | 14.3±0.0 | 79.3±1.0 | 15.1±0.8 | 78.4±1.5 | 91.3±1.0 | 75.7±1.0 | 83.6±0.3 | 83.8±0.6 | **93.2±0.8** | 82.9±0.3 |
| | | Prune+LD | 78.4 | 14.3±0.0 | 76.7±2.2 | 15.2±1.1 | 77.8±1.0 | 87.6±2.2 | 71.5±1.7 | 80.4±1.7 | 80.1±0.8 | **93.6±0.2** | 80.4±0.1 |
| | | OD | 83.9 | 14.3±0.0 | 81.6±0.3 | 65.0±24.1 | 80.9±3.8 | 92.3±0.7 | 80.5±3.4 | 86.7±0.4 | 83.3±1.9 | **97.7±0.4** | 81.2±0.6 |
| Pubmed | 150 | None | 84.1 | 33.3±0.0 | 86.1±0.3 | 82.8±2.7 | 85.6±0.1 | 79.1±1.1 | 85.6±0.2 | 88.5±0.3 | 86.6±0.2 | **96.0±0.1** | 84.9±0.3 |
| | | Prune | 85.0 | 33.3±0.0 | 86.6±0.1 | 33.4±0.0 | 86.2±0.2 | 85.2±2.2 | 85.5±0.2 | 84.8±0.3 | 86.4±0.2 | **94.9±0.1** | 85.3±0.5 |
| | | Prune+LD | 85.1 | 33.3±0.0 | 83.9±0.2 | 33.4±0.1 | 83.7±0.2 | 70.5±0.8 | 84.0±0.3 | 88.8±0.2 | 85.7±0.1 | **94.5±0.3** | 84.7±0.3 |
| | | OD | 85.1 | 33.3±0.0 | 86.2±0.1 | 84.5±5.9 | 86.1±0.3 | 83.5±1.5 | 86.5±0.2 | 85.8±0.2 | 86.4±0.2 | **95.5±0.4** | 85.6±0.1 |
| Bitcoin | 300 | None | 78.3 | 33.3±0.0 | 78.3±0.0 | 84.3±12.7 | 78.3±0.0 | 88.0±17.8 | 78.3±0.0 | 80.8±7.0 | 78.3±0.0 | **99.3±0.1** | 78.3±0.0 |
| | | Prune | 78.3 | 33.3±0.0 | 78.3±0.0 | 35.2±3.9 | 78.3±0.0 | 81.6±9.6 | 78.3±0.0 | 33.3±0.0 | 78.3±0.0 | **99.1±0.1** | 78.3±0.0 |
| | | Prune+LD | 78.3 | 33.3±0.0 | 78.3±0.0 | 40.1±10.5 | 78.3±0.0 | 80.6±12.6 | 78.3±0.0 | 33.3±0.0 | 78.3±0.0 | **99.2±0.1** | 78.3±0.0 |
| | | OD | 78.3 | 33.3±0.0 | 78.3±0.0 | 44.7±11.4 | 78.3±0.0 | 33.3±0.0 | 78.3±0.0 | 88.0±0.2 | 78.3±0.0 | **99.3±0.1** | 78.3±0.0 |
| Facebook | 100 | None | 86.0 | 25.0±0.0 | 86.8±0.2 | 81.9±5.8 | 86.6±0.3 | 85.6±0.8 | 86.7±0.3 | 85.2±0.3 | 87.1±0.2 | **86.9±0.3** | 86.0±0.3 |
| | | Prune | 85.9 | 25.0±0.0 | 86.7±0.4 | 25.3±0.5 | 86.6±0.2 | 85.8±1.1 | 86.3±0.2 | 70.0±0.1 | 86.9±0.2 | **87.0±0.2** | 86.0±0.2 |
| | | Prune+LD | 85.1 | 25.0±0.0 | 86.6±0.2 | 25.3±0.3 | 86.2±0.2 | 77.9±0.8 | 85.8±0.4 | 73.3±0.2 | 86.9±0.4 | **87.3±0.4** | 86.1±0.2 |
| | | OD | 85.6 | 25.0±0.0 | 86.8±0.2 | 76.3±4.6 | 86.1±0.4 | 72.4±2.8 | 86.9±0.3 | 84.7±0.2 | 87.1±0.7 | **87.1±0.5** | 86.4±0.5 |
| Flickr | 300 | None | 45.0 | 14.3±0.0 | 47.6±0.3 | **99.7±0.2** | 45.4±0.6 | 28.9±0.9 | 46.1±0.5 | 54.3±0.3 | 47.0±0.2 | 94.5±1.0 | 44.9±0.4 |
| | | Prune | 45.0 | 14.3±0.0 | 46.0±0.1 | 15.1±0.4 | 43.6±0.2 | 34.7±2.2 | 45.3±0.3 | 66.9±0.7 | 46.7±0.4 | **95.0±1.4** | 44.6±0.5 |
| | | Prune+LD | 45.0 | 14.3±0.0 | 44.3±0.4 | 14.8±0.2 | 43.7±0.2 | 28.2±0.3 | 45.3±0.5 | 20.9±2.7 | 45.8±0.5 | **95.6±1.2** | 45.6±0.2 |
| | | OD | 45.6 | 14.3±0.0 | 47.1±0.3 | 24.8±5.5 | 44.4±1.1 | 18.8±0.6 | 46.5±0.3 | 85.0±0.0 | 46.2±0.3 | **94.3±2.1** | 46.4±0.1 |
| OGB-arxiv | 800 | None | 64.8 | 2.5±0.0 | 66.5±0.4 | 68.4±1.9 | 66.1±0.6 | 43.4±2.2 | 64.9±0.8 | 73.3±0.1 | 65.4±0.2 | **74.8±0.6** | 65.7±0.2 |
| | | Prune | 65.0 | 2.5±0.0 | 65.1±0.4 | 2.9±0.5 | 64.7±0.5 | 54.3±0.5 | 65.3±0.5 | 13.1±0.8 | 65.5±0.1 | **74.9±0.4** | 66.1±0.3 |
| | | Prune+LD | 63.2 | 2.5±0.0 | 64.5±0.1 | 3.3±0.6 | 64.8±0.4 | 53.3±1.4 | 65.0±0.9 | 5.7±0.6 | 65.1±0.2 | **74.3±0.3** | 65.6±0.2 |
| | | OD | 64.9 | 2.5±0.0 | 64.7±0.2 | 24.6±0.9 | 63.7±0.5 | 38.1±1.5 | 65.8±0.2 | 75.1±0.2 | 65.9±0.1 | **73.9±0.9** | 65.8±0.6 |

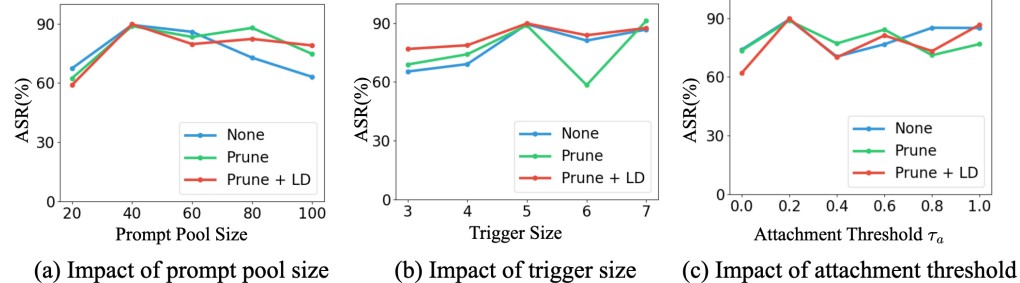

(a) Impact of prompt pool size     (b) Impact of trigger size     (c) Impact of attachment threshold

Figure 4: The comparison of different (a) trigger pool size, (b) trigger size, and (c) attachment threshold $\tau_a$ on Flickr dataset.

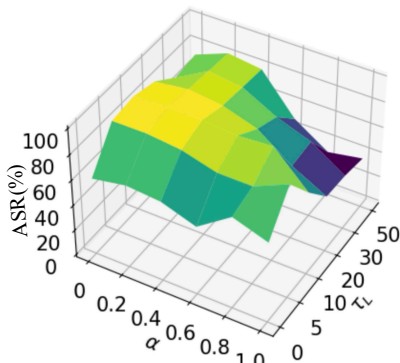

Figure 5: The comparison of different $\alpha$ and $\tau_L$ on Flickr dataset.

results show that EUMC achieves the best ASR among different defense strategies with similar training times as other methods.

## D  Detailed Experiments on GNNs

In Tables 5, 6, and 7, we present detailed experimental results, including the average and standard deviation of ASR and clean accuracy, on various GNN architectures, namely GCN [16], GAT [23], and GraphSAGE [26]. From these tables, we can find that EUMC method achieves state-of-the-art results on all datasets except for Pubmed, demonstrating its robustness and generalization ability across different GNN structures for multi-category graph backdoor attacks. The comparable performance of EUMC to UGBA on Pubmed likely stems from the dataset's small size and limited number of categories (only three), which constrain the effectiveness of attacks. Moreover, when analyzing performance variations across different GNN architectures on the same dataset, our model shows a balanced performance, indicative of a strong generalization ability of our graph backdoor attack method. Specifically, on datasets like Flickr and OGB-arxiv, although there is a noticeable performance difference between GAT and GCN, our model maintains a better balance compared to other methods. This consistency highlights the unique advantages of EUMC in managing multi-category graph backdoor attacks, emphasizing its potential for widespread application across diverse settings.

In these tables, we also notice that the performance of SBA is relative poor due to two reasons, 1) the smaller poison node set; 2) the multi-category graph backdoor attack setting. Specifically, in UGBA [13] and DPGBA [14], SBA has already performed poorly in single-category graph backdoor attack on node classification when the size of poison node set gets small. Moreover, our work focuses on more challenging multi-category graph backdoor attack, where the different triggers are required to capture different category-aware feature. Therefore, the performance of SBA get further reduced.

# E Impacts of Trigger Pool Size

The subgraph triggers pool offers various attack patterns for the backdoor attack. To examine the impact of these patterns, we vary the trigger pool size for each category as {20, 40, 60, 80, 100} and plot the average ASR for GCN, GAT, and GraphSAGE with different defense strategies in Figure 4 (a). From this figure, we observe that as the trigger pool size increases from 20 to 100, the performance of EUMC method first increases, peaking at a pool size of 40, and then decreases. A smaller trigger pool size limits the attack patterns available for each target category, which undermines the performance of our model. Conversely, a larger trigger pool size can include subgraph triggers that are not well-optimized, which also affects the performance of the graph backdoor attack.

# F Impacts of Trigger Size

To examine the impact of different trigger sizes, we conduct experiments to explore the attack performance of our method by attaching varying numbers of nodes as triggers for a poisoned node. Specifically, we vary the trigger size in increments {3, 4, 5, 6, 7}, and plot the average ASR for GCN, GAT, and GraphSAGE on Flickr with different defense strategies in Figure 4 (b). From the experimental results, we observe that as trigger size increases, the attack success rate initially increases and then stabilizes. Given that including more nodes in each trigger could potentially expose our attack, we opt to construct each trigger with five nodes to mislead the backdoored model.

# G Impacts of Attachment Threshold $\tau_a$

The attachment threshold $\tau_a$ strikes a balance between attack intensity and unnoticeability. To investigate the effects of $\tau_a$, we adjust its values in increments 0, 0.2, 0.4, 0.6, 0.8 and plot the average Attack Success Rate (ASR) for GCN, GAT, and GraphSAGE on Flickr with various defense strategies in Figure 4 (c). The experimental results reveal that as $\tau_a$ increases, the average ASR initially rises and then declines. When $\tau_a$ is set low, more nodes from the subgraph trigger can be attached to the attacked node, potentially harming the generative capability and making it more detectable by defense algorithms. Conversely, as $\tau_a$ increases, fewer nodes from the subgraph trigger are attached to the attacked node, which may limit the effectiveness of graph backdoor attack.

# H Impacts of Similarity Loss $\mathcal{L}_h$

To examine the effect of similarity between subgraph triggers and attacked nodes, we investigate how the hyper-parameters $\alpha$ and $\tau_L$ influence the performance of EUMC. Here, $\alpha$ controls the weight of $\mathcal{L}_h$, and $\tau_L$ determines the threshold for similarity scores used in $\mathcal{L}_h$. We vary $\alpha$ values as {0, 5, 10, 20, 30, 50} and $\tau_L$ from {0, 0.2, 0.4, 0.6, 0.8, 1}, and plot the average ASR for GCN, GAT, and GraphSAGE on Flickr with different defense strategies in Figure 5. For both $\alpha$ and $\tau_L$, the ASR initially increases and then decreases, indicating that setting the similarity between trigger nodes and attacked nodes, based on the node similarity distribution of the original graph, can make the attack unnoticeable. However, overly emphasizing this similarity can also compromise the effectiveness of the subgraph triggers.

