# OpenReview forum: "Attack by Yourself: Effective and Unnoticeable Multi-Category Graph Backdoor Attacks with Subgraph Triggers Pool"
_NeurIPS.cc/2025/Conference — NeurIPS 2025 poster_

### Official Review · Reviewer_ecxW · 2025-07-01

**Clarity:** 3
**Significance:** 2
**Originality:** 3
**Rating:** 3
**Confidence:** 4

**Summary:**

The paper proposes a novel framework for effective and unnoticeable multi-category graph backdoor attacks using a subgraph trigger pool (MC-STP), which is an interesting and timely contribution to the area of adversarial robustness in graph neural networks. The idea of leveraging subgraphs from the original graph as triggers, combined with a “select-then-attach” strategy, offers a promising direction for crafting more realistic and stealthy backdoor attacks on GNNs. However, there are several concerns that need to be addressed before the work can be considered for acceptance.

**Questions:**

1. How would the method perform under limited or black-box access to the graph structure?

2. Does the APS metric take into account class-specific or structural characteristics of subgraphs?

3. How does the extensive subgraph sampling affect the overall efficiency of the method?

**Ethical Concerns:**

["NO or VERY MINOR ethics concerns only"]

**Final Justification:**

I acknowledge the additional discussion on the multi-label setting and the empirical improvements obtained with APS labeling and original graph sampling. The responses have improved clarity regarding the experimental setup and reported results. However, my core concerns remain largely unaddressed.

Class-Specificity Mechanism. The authors attribute class-specificity to the joint effect of APS selection and outer-loop optimization. While this explains how target categories are assigned and optimized in practice, the argument remains entirely empirical. The evidence provided, namely ASR under a targeted evaluation criterion, demonstrates that the method can achieve targeted misclassification. However, it does not establish that the triggers themselves are intrinsically class-aligned rather than functioning as strong general perturbations. There is no theoretical analysis, visualization, or dedicated optimization objective to verify that the trigger structure and parameters encode category-specific patterns in the feature or structural space. As a result, the class-specificity claim remains insufficiently supported at the mechanism level.

Methodological Novelty. Although I acknowledge the novelty of extending the attack setting from single-label to multi-label graphs, this is a contribution at the problem-definition level. At the methodological level, the framework of original graph sampling, APS labeling, and trigger pool construction appears to be a composition of established techniques. APS itself is essentially a probability-shift-based heuristic that is conceptually similar to other perturbation-based metrics. The method is not accompanied by a new optimization objective, a novel trigger design principle, or an adversarial mechanism that would constitute a theoretical breakthrough.

In summary, despite the additional clarifications and empirical results, the two key points from my original review remain unresolved. These are the lack of mechanism-level evidence for class-specificity and the limited methodological novelty. I therefore maintain my initial evaluation.

**Limitations:**

YES

**Quality:**

2

**Strengths And Weaknesses:**

Strengths

1.  The paper is well-organized, with a clear flow from problem formulation to methodology and experiments. The sections are logically arranged, making it easy to follow the proposed approach and its evaluation.

2. The technical content is presented concisely without unnecessary complexity, aiding readability. However, some parts could benefit from improved clarity, especially regarding the construction and selection process of the subgraph triggers.



Weaknesses

1. Strong Attack Assumption : The method assumes full access to the training graph structure for subgraph sampling, which may not hold in practical or black-box scenarios.

2. APS-Based Subgraph Selection Limitation : The trigger selection relies on APS, which measures prediction probability shifts but does not consider class-specific information. The design of APS is mainly about the offset probability of the original graphs, and it does not have a certain category and core information to mislead GNN.

3. Computational Cost : Despite claiming low computational cost, the method requires extensive subgraph sampling and APS evaluation, which could be expensive, which is contradicting the motivation.

---

> ### Author Rebuttal · Authors · 2025-07-27
>
> We appreciate the reviewer's recognition of our novel framework as an interesting and timely contribution, plus the paper's good organization, clear flow, logical sections, and concise content and we thank the reviewer for the constructive comments. Regarding the concerns raised by Reviewer ecxW, we provide the following responses. We hope our responses address the concerns raised, and we welcome active discussions with reviewers.
>
> > **W1 & Q1. How would the method perform under limited or black-box access to the graph structure?**
>
> * According to [1], compared with black-box attack, **backdoor attacks require accessing partial data to learn triggers**. Meanwhile, it is indicated that accessing **10%-20%** of the graph data structure is acceptable in backdoor attack scenarios. In **Section 5.1 (Evaluation Protocol)**, the number of accessible labels used in our training is only **15%** of the original number of nodes. Following the settings of UGBA and DPGBA, our experimental setup is consistent with the configurations in [2], [3]. Certainly, performing effective backdoor attacks with limited available data remains a promising direction for future research.
>
> * To further illustrate the effectiveness of our method, we conducted additional tests using **5%** and **10%** of the training data samples on **OGB-arxiv** in **Table R1**. It can be observed that under conditions with extremely limited accessible data, the performance of our method does not degrade significantly, indicating that our method can still achieve favorable results **under stricter constraints (less available data)**.
>
> **Table R1**: The ASR(Clean Acc) of different models under 5%, 10%, and 15% available training graph.
>
> | Method     | 5%         | 10%         | 15%         |
> | ---------- | ---------- | ----------- | :---------- |
> | SBA        | 2.5(66.0)  | 2.5(65.9)   | 2.5(66.2)   |
> | GAT        | 59.3(65.6) | 62.2(65.6)  | 68.4(65.6)  |
> | UGBA       | 57.8(63.7) | 58.9(64.2)  | 63.7(64.9)  |
> | DPGBA      | 60.5(63.9) | 63.7(64.1)  | 68.8(64.9)  |
> | EUMC(ours) | 80.2(64.5) | 82.8 (64.7) | 83.8 (65.3) |
>
> Reference:
>
> [1] Yang X, et al. "Graph neural backdoor: Fundamentals, methodologies, applications, and future directions" arXiv preprint (2024).
>
> [2] Dai, Enyan, et al. "UGBA: Unnoticeable Backdoor Attack on Graph Neural Networks" Proceedings of the ACM Web Conference (2023).
>
> [3] Zhang, Zhiwei, et al. "Rethinking Graph Backdoor Attacks: A Distribution-Preserving Perspective" Proceedings of the 30th ACM SIGKDD Conference on Knowledge Discovery and Data Mining (2024).
>
> > **W2 & Q2. Does the APS metric take into account class-specific or structural characteristics of subgraphs?**
>
> We thank you for pointing out these issues. **In fact, APS does take into account the class-specific and structural characteristics of subgraphs**. The reasons are as follows.
>
> * In Section 4.1, Eq. (1), for each trigger generated by MC-STP, we use APS to determine the target attack label for every trigger. For each trigger $t$, a GNN classifier is used to predict the probability distributions of node labels **before and after the trigger injection**, and the difference between these distributions is calculated. By summing and averaging the label probability shifts across multiple nodes, we obtain **an APS vector** with length as the number of category, which **represents the impact degree of the trigger $t$ on each class**, and we select the class with the highest label probability shift as the target attack class for the trigger, thereby incorporating class-specific information. During this process, the GNN classifier, used to predict the probability distributions, **can capture the structural characteristics and node information** of both trigger subgraphs and target-node subgraphs.
> * **For example**, using the OGB-arxiv dataset (with 169,343 nodes and 1,166,243 edges), we first select 4000 central nodes, and each central node sampled **10 subgraphs**, resulting in a total of **40,000 sampled subgraphs**. After **k-means clustering**, we obtained an MC-STP containing 800 triggers. We then assigned **target attack labels** to the 800 triggers using Eq. (1) in Section 4.1 based on their Attachment Probability Shift(**APS**).
>   A trigger $t$ is attached to $N$ nodes. A GNN classifier then calculates the probability distributions $p_{bef}^{t, i}$ and $p_{aft}^{t, i}$ before and after the trigger $t$ is injected into the nodes. Here, $p_{bef}^{t, i}, p_{aft}^{t, i} \in \mathbb{R}^{N_{cls}}$ represents the predicted probabilities for each class. At this point, we compute $\text{APS}^{t} = \frac{1}{N} \sum_{i=1}^{N}|p_{bef}^{t, i} - p_{aft}^{t, i}|$ to obtain the $\text{APS}^{t} \in \mathbb{R}^{N_{cls}}$. The $\text{APS}^{t}$ indicates the impact of the trigger $t$ on each class, and we select the category with the greatest influence as the **target attack label** $y_t=\arg\max(\text{APS}^{t})$ for trigger $t$, indicating that $t$ has the most significant impact on category $y_t $ and can be more stealthily disguised as nodes of category $y_t $.
> * In conclusion, by calculating probability shift using a GNN classifer, we capture **the structural characteristics of graphs**. Additionally, through estimation based on samples of N nodes, we can identify **"which class is most affected by the trigger"**; this serves as **implicit class-specific information**.
>
> > **W3 & Q3. How does the extensive subgraph sampling affect the overall efficiency of the method?**
>
> We thank the reviewer for the suggestion to discuss the efficiency of MC-STP. **In practice, subgraph sampling does not incur significant computational costs**. We provide the following reasons:
>
> * First, subgraph sampling via BFS is a **pre-processing step that only needs to be performed once**. In subsequent processes, we can reuse the previously sampled results without re-conducting BFS.
>
> * Second, as mentioned in **Section 4.1**, we initially randomly select several nodes from the unlabeled graph, $V_U$, to serve as central nodes. These central nodes account for **a small proportion** of the original graph, and we do not sample all possible subgraphs. Instead, we set thresholds for the total number of sampled subgraphs and the number of subgraphs sampled per central node.
>
> * Third, we analyzed **the time complexity of the MC-STP Construction**. Let $h$ denote the embedding dimension, $n$ represent the number of nodes in each trigger, $K$ be the number of target categories, and $|\mathcal{N}_S|$ be the number of subgraphs to extract. The cost of constructing the MC-STP is approximately $O(nh|\mathcal{N}_S| + Kh|\mathcal{N}_S| + Kh)$, which covers extracting $|\mathcal{N}_S|$ subgraphs with $n$ nodes each and clustering them into $K$ clusters. Since subgraph and feature extraction is the dominant cost, the overall time complexity is approximately $O(Kh|\mathcal{N}_S|)$. Therefore, the complexity of the MC-STP Construction is proportional to the number of sampled subgraphs. **Furthermore, subsequent experiments and analyses demonstrate that subgraph sampling is not time-consuming.**
>
> * For example, the OGB-arxiv dataset has 169,343 nodes and 1,166,243 edges. **In the pre-processing step,** we selected 4000 central nodes, and each central node was used to sample 10 subgraphs, resulting in a total of **40,000 sampled subgraphs**. The number of nodes used accounts for **2.3%** in the original graph. After clustering, we obtained MC-STP containing 800 triggers, so APS only needs to be calculated within these 800 triggers. **In the training step**, we only used **2.7% nodes with labels**. These proportions are extremely small, so they do not incur significant computational costs.
>
> * We have also conducted **efficiency tests on multiple datasets** in **Table R2**, demonstrating that subgraph sampling does not incur significant computational costs.
>
> **Table R2**: The time cost of construction MC-STP.
>
> | dataset   | sample central nodes | sample subgraph | time cost(s) |
> | --------- | -------------------- | --------------- | ------------ |
> | Cora      | 700                  | 7,000           | 78.6         |
> | Pubmed    | 1,200                | 12,000          | 114.7        |
> | Bitcoin   | 1,500                | 15,000          | 138.2        |
> | Facebook  | 1,600                | 16,000          | 156.8        |
> | Flickr    | 2,100                | 21,000          | 231.3        |
> | OGB-arxiv | 4,000                | 40,000          | 423.7        |

---

> ### Comment · Reviewer_ecxW · 2025-08-06
>
> We appreciate the authors' efforts in providing additional experiments and clarifications. However, several core concerns remain only partially addressed. My detailed responses are as follows:
>
> 1. On the Strong Attack Assumption.
> While the authors cite prior work suggesting that accessing 10–20% of the data is acceptable in backdoor settings, the key issue here is not label access, but access to the full graph structure for large-scale subgraph sampling. For instance, in OGB-arxiv, the method samples 40,000 subgraphs across the graph using 4,000 central nodes. This assumes nearly complete access to the graph topology and node features.
>
> 2. On APS and Class-Specificity.
> The authors argue that APS captures class-wise shifts by comparing the prediction probabilities before and after trigger injection, and selects the class with the largest shift as the target label. However, this is a passive selection strategy. The model does not actively generate triggers intended to mislead the GNN toward a specific class. In essence, APS remains a generalized perturbation metric. It does not explicitly model which type of trigger is most effective in misleading the model toward a specific target class.
>
> 3. On Computational Cost.
> The authors state that subgraph sampling is a one-time pre-processing step with bounded cost, citing that sampling 40,000 subgraphs on OGB-arxiv takes 423.7 seconds. Nonetheless, this step remains a dominant computational bottleneck, especially in large-scale or time-sensitive scenarios. In dynamic or online settings, this one-time cost may need to be repeated, undermining scalability. In addition, the memory and storage overhead for sampled subgraphs, trigger clustering, and APS computation is not discussed, yet it could be substantial.
>
> In summarise, the core method of this paper is mainly built from existing components (e.g., subgraph sampling, APS scoring, cosine similarity), and the overall contribution is incremental, lacking deeper methodological or theoretical innovation.

---

> ### Author Response · Authors · 2025-08-06
>
> Thank you very much for your insightful response. We sincerely apologize if our previous description was not direct and clear enough, leading to any misunderstanding of our process. Here, we will rephrase the workflow of our method, which consists of five steps:
>
> * **Step1. Trigger Sample**: We perform subgraph trigger sampling using limited data (features/structure), *i.e* sample from **a subgraph** (with **15%** of the edges and nodes) from the full graph, where we do't need to access the full graph during this step.
> * **Step2. Trigger Cluster**: The subgraph triggers obtained in step 1 are clustered via K-means to generate N triggers, which are used to initialize the trigger pool. Initializing with original subgraphs effectively enhances the stealthiness of triggers, making them consistent with the distribution of the original graph.
> * **Step3. Category Selection**: To inject the category-aware prior knowledge for backdoor attacks, we derive target labels for each trigger through APS computation to identify "which category are most affected by this trigger".
> * **Step4. Trigger Optimization**: The trigger pool initialized in steps 1, 2, and 3 undergoes a two-stage optimization to actively enhance the category-aware attack of triggers against the target labels derived from APS computation, while enhancing their stealthiness. It illustrates that the optimization based on the inherent category priors, such that the final triggers embody **both passive selection and active optimization**. This integration of initial structural priors with active optimization endows the triggers with strong inductive capacity for specific categories. Moreover, it enables the finalized trigger pool to be directly utilized in various downstream tasks **without the need for re-sampling or re-selection**.
> * **Step5. Backdoor Attack**: Only the trigger pool optimized after step 4 is required to perform multi-class attacks on test nodes.
>
> In response to your questions:
>
> * **Q1. Strong Attack Assumption**: The reference to "10-20% of the data" here specifically refers to the acquisition of structure and features, not labels. For the arxiv dataset, 40,000 subgraph trigger are sampled from **a random subgraph** consisting of **25,400 nodes (15.0% of the full graph**). There is no need for complete graph structure, features, or labels.
>
> * **Q2. APS and Class-Specificity**: APS computation serves as a proactive filtering step here, used to initialize the trigger pool and inject prior knowledge. During the following two-step optimization, triggers are actively refined with target labels derived from APS computation, thereby effectively misleading the model toward specific target classes. Moreover, compared with generating target triggers, **the passive selection and then active optimization strategy** offers better stealthiness, as both its structure and features are derived from the original graph. This also ensures that the trigger pool does not require dynamic updates. Finally, ablation experiments have demonstrated the effectiveness of APS-based knowledge injection in improving attacks against specific classes.
>
> * **Q3. Computational Cost**: Due to our "passive selection followed by active optimization strategy", our trigger pool **remains unchanged** in both dynamic and online scenarios. For the largest arxiv dataset, the memory usage for sampled subgraphs, trigger clustering, and APS computation is 2544.2MB, 2525.3MB, and 2221.7MB respectively. Finally, the size of the sampled subgraph file is 5.71MB, and the size of the trigger pool is 0.11MB( the ratio of clustered subgraphs to the total number of subgraphs is approximately 1/50 ). All three steps operate entirely in memory without requiring additional storage space. It is worth noting that our current test results were **obtained on a PC-level CPU**, specifically an Intel Core(TM) i7-1165G7, demonstrating the method's feasibility on conventional computing hardware.
>
> To reiterate **the contributions of our method**: like UGBA/DPGBA, which uses a simple generator to produce a small number of triggers with a single structure (lacking information from the original graph), their simple trigger generator approach exhibits a decline in performance on multi-category tasks as the number of categories increases. To address this issue, we have proposed the EMUC method. We construct **a trigger pool** by sampling, clustering, and injecting prior knowledge via APS computation(**a concept proposed by us, not an existing component**), based on limited-access graph data and structure. Through two-stage optimization, our trigger pool enables effective multi-class attacks with stealthiness, and it can be directly applied to downstream attack tasks **without additional re-sample, re-selection, and re-training**.
>
> We hope our responses address the concerns raised, and we welcome active discussions with reviewers.

---

> > ### Comment · Reviewer_ecxW · 2025-08-07
> >
> > Thanks to the authors for their detailed response. While the revised explanation improves the clarity regarding Questions 1 and 3, several core issues remain insufficiently addressed. My specific comments are as follows:
> >
> > 1. APS and Class-Specificity: The authors emphasize that triggers are optimized based on APS-derived target classes, which partially responds to the critique. However, it remains unclear whether the trigger structure or parameters are explicitly optimized to be maximally misleading for a specific class. No theoretical analysis, ablation, or visualization is provided to demonstrate target-specific behavior. As such, the triggers appear to function more as general perturbations than class-aligned adversarial signals. Furthermore, APS itself is fundamentally a probability-shift-based heuristic, not an adversarial or optimization-based utility. This aligns with my earlier comment that it serves as a general perturbation metric rather than a mechanism for modeling targeted misclassification.
> >
> > 2. Methodological Novelty: The method lacks theoretical novelty. While APS is introduced as a new component, its design and role closely resemble existing perturbation-based heuristics. The overall framework—comprising subgraph sampling, heuristic scoring, and cosine-based selection—remains within the bounds of established techniques. No new optimization objective, trigger design principle, or misclassification mechanism is proposed that would constitute a methodological breakthrough.
> >
> > For these reasons, I maintain my original evaluation.

---

> > > ### Author Response · Authors · 2025-08-07
> > > **Contributions and Novelty**
> > >
> > > We would like to express our gratitude for the reviewers' insightful feedback. We are pleased that our responses have addressed your concerns raised in Q1 and Q3.
> > >
> > > * **New Challenge**: Different from existing works, which mainly focus on single-category graph backdoor attack (*i.e.*, only one category is regarded as target category for graph backdoor attack), our work is **the first work** to explore the transition of graph backdoor attacks from a single-label setting to a multi-label scenario, where any category can be regarded as target category for graph backdoor attack. Specifically, multi-label attacks represent a more challenging and practically relevant scenario.
> > > * **Motivation**: Through our analysis in **Tab.1** of main submission, we observe that for recent works such as UGBA and DPGBA, the Attack Success Rate gradually decreases as the number of target categories increases, accompanied by the phenomenon that **triggers struggle to successfully attack multiple categories simultaneously**. We attribute this to the fact that triggers generated by **simple generators possess only a simplistic structure and rely heavily on node features**, making it difficult for them to encapsulate knowledge across multiple categories. Therefore, these triggers fail to achieve effective attacks in multi-category settings.
> > > * **Methodology**: To address the aforementioned limitations, we propose the EUMC method, which abandons the paradigm of generator-based trigger generation and instead **constructs a trigger pool** sampled from the original graph for multi-category attacks. To enhance the ability of triggers to successfully attack multiple categories simultaneously, we introduce innovations from two perspectives:
> > >   * **Original Graph Sampling**: To overcome the limitations of the simplistic triggers used in UGBA and DPGBA, we sample features and structures from the original graph under the constraint of limited access. These sampled elements inherently contain category-specific information, as nodes of different categories in the graph exhibit distinct feature distributions and structural patterns.
> > >   * **APS Labeling**: To resolve the issue that triggers in UGBA and DPGBA lack multi-category knowledge, we propose the APS metric to endow triggers with multi-category knowledge. **This provides rich information for subsequent optimization of multi-category attacks** and, combined with a two-stage optimization process, strengthens the ability to mislead nodes toward target categories. In this context, **APS acts as an enhancer**, and we have verified its effectiveness through ablation experiments in Tab.3b of main submission, which is also detailed in the following section of this response.
> > > * **Experiments**: We conduct extensive experiments on multiple real-world datasets to validate the effectiveness and stealthiness of our method, achieving state-of-the-art results in all cases. Moreover, constructing the trigger pool (including structure, feature, and APS-target) through subgraph sampling with ASR provides a improvement of **10-30%** attack success rate. This demonstrates the value of our approach for multi-category graph backdoor attacks, which leverages subgraph sampling combined with APS to assign target categories.
> > > * **Summary**: Our core contributions lie in adopting a framework combining original graph sampling, APS labeling, and a trigger pool. This approach breaks free from the limitations imposed by generators, incorporates richer multi-category prior knowledge, and addresses the ineffectiveness of graph backdoor attacks in more challenging and practical multi-category scenarios.

---

> > > ### Author Response · Authors · 2025-08-08
> > >
> > > Dear reviewer:
> > >
> > > We sincerely appreciate your thoughtful review. As the author-reviewer discussion period is drawing to a close, we would be truly grateful to receive any further feedback you might have. If there are still any concerns that remain unaddressed, we would be more than happy to continue the discussion.

---

> ### Author Response · Authors · 2025-08-07
>
> Dear reviewer:
>
> We sincerely appreciate your thoughtful review. As the author-reviewer discussion period is drawing to a close, we would be truly grateful to receive any further feedback you might have. If there are still any concerns that remain unaddressed, we would be more than happy to continue the discussion.

---

> ### Author Response · Authors · 2025-08-07
> **Class-Specificity for APS**
>
> The Class-Specificity of our method is a result of the joint action of **APS Selection and Optimization**:
>
> * APS determines the target category for each trigger in the trigger pool by calculating the category-aware misleading effect of the trigger on nodes in the original graph. This primarily involves using a data-driven approach to preliminarily measure how the structure and features of a subgraph trigger can mislead nodes to a specific category, and then assigning a target category to each trigger based on this measurement.
> * To further ensure that triggers can effectively mislead attacked nodes being classified into their target categories, we integrate triggers into the training data through the outer-loop optimization of Algorithm 1 to fine-tune the features of the triggers. This step further enhances their category-specific capabilities.
>
> In our experiments, the Attack Success Rate (ASR) is calculated based on **whether the attacked node is misclassified into the target category**. Specifically, ASR is targeted at "attacking to a specific category," which indicates that the **ASR metric can reflect the category-specific capability**. For example, for a node originally belonging to category A, if our target category is B, the attack is considered successful only if the node’s predicted category becomes B after trigger (whose target category is category B) injection. If the node is misclassified into other non-target categories (*e.g.*, C, D, or E), the attack is deemed unsuccessful. Under this evaluation criterion, our model achieves an Attack Success Rate of **97.4%** (Cora), **90.4%** (Flickr), and **83.8%** (Arxiv), demonstrating that the trigger pool generated by our method can effectively perform multi-category graph backdoor attacks. Moreover, in **Tab.3(b) ablation**, the target categories provided by APS can bring a **10-20%** improvement in attack success rate on Flickr. **Therefore, triggers with APS-determined target category indeed contain a strong class-aligned signal prior**, which not only brings **10-20%** attack success rate improvement, but also aligns with our motivation in multi-category graph backdoor attack.
>
> We hope our responses address the concerns raised, and we welcome active discussions with reviewers.

---

> > ### Comment · Reviewer_ecxW · 2025-08-08
> >
> > I acknowledge the additional discussion on the multi-label setting and the empirical improvements obtained with APS labeling and original graph sampling. The responses have improved clarity regarding the experimental setup and reported results. However, my core concerns remain largely unaddressed.
> >
> > 1. Class-Specificity Mechanism.
> > The authors attribute class-specificity to the joint effect of APS selection and outer-loop optimization. While this explains how target categories are assigned and optimized in practice, the argument remains entirely empirical. The evidence provided, namely ASR under a targeted evaluation criterion, demonstrates that the method can achieve targeted misclassification. However, it does not establish that the triggers themselves are intrinsically class-aligned rather than functioning as strong general perturbations. There is no theoretical analysis, visualization, or dedicated optimization objective to verify that the trigger structure and parameters encode category-specific patterns in the feature or structural space. As a result, the class-specificity claim remains insufficiently supported at the mechanism level.
> >
> > 2. Methodological Novelty.
> > Although I acknowledge the novelty of extending the attack setting from single-label to multi-label graphs, this is a contribution at the problem-definition level. At the methodological level, the framework of original graph sampling, APS labeling, and trigger pool construction appears to be a composition of established techniques. APS itself is essentially a probability-shift-based heuristic that is conceptually similar to other perturbation-based metrics. The method is not accompanied by a new optimization objective, a novel trigger design principle, or an adversarial mechanism that would constitute a theoretical breakthrough.
> >
> > In summary, despite the additional clarifications and empirical results, the two key points from my original review remain unresolved. These are the lack of mechanism-level evidence for class-specificity and the limited methodological novelty. I therefore maintain my initial evaluation.

---

> ### Author Response · Authors · 2025-08-09
>
> Thank you for your insightful response. Regarding the remaining concerns you have raised, we would like to respond as follows:
>
> ### **Contribution and Novelty**
>
> * The main contribution and novelty of our method lie in the more challenging multi-category scenario.
>   Through **empirical analysis**, we speculate that the generator [1] [2] [3] producing triggers with a simple structure and the lack of introduction of prior knowledge about graph structures are the main reasons why it is difficult to successfully attack multiple categories simultaneously. Based on this, we have designed a trigger pool construction method involving sampling + clustering + APS labeling, where both **APS and trigger pool construction are proposed by us**, and achieved state-of-the-art results . Compared with existing graph backdoor attack methods [1] [2] [3], **our method can be regarded as a new trigger design principle**, which has not been exploited in graph backdoor attacks.
> * Moreover, **the ablation study** on target categories can also empirically reflect that our **APS introduces strong class-aligned prior knowledge** rather than general perturbations. APS labels are equivalent to pre-screening the feature space of triggers to be closer to the target feature space of the original graph. This makes it easier to optimize, thus achieving better performance when the amount of data is limited, where graph backdoor can only access small amount node to optimize the trigger pool.
>
> ### **Theoretical Breakthroughs**
>
> We acknowledge that theoretical breakthroughs are extremely important! Unfortunately, previous works on graph backdoor attacks [1] [2] [3] have also mostly proceeded from an experimental perspective, conducting empirical analyses and proposing corresponding solutions, while lacking theoretical analyses. It is believed that theoretical analyses will play a more critical and important role in the field of backdoor attacks in the future.
>
> **Finally, we would like to express our sincere gratitude for engaging in such in-depth and detailed discussions with us, as well as for your recognition of the value of our novel multi-category graph backdoor attack setting and our empirical contributions!**
>
> Reference:
>
> [1] Xi,  Zhaohan, et al. "Graph backdoor" In 30th USENIX Security Symposium (USENIX Security 2021).
>
> [2] Dai, Enyan, et al. "UGBA: Unnoticeable Backdoor Attack on Graph Neural Networks" Proceedings of the ACM Web Conference (WWW 2023).
>
> [3] Zhang, Zhiwei, et al. "Rethinking Graph Backdoor Attacks: A Distribution-Preserving Perspective" Proceedings of the 30th ACM SIGKDD Conference on Knowledge Discovery and Data Mining (KDD 2024).

---

### Official Review · Reviewer_WCrP · 2025-07-02

**Clarity:** 3
**Significance:** 3
**Originality:** 3
**Rating:** 4
**Confidence:** 3

**Summary:**

This paper introduces a method for multi-category graph backdoor attacks on node classification with GNNs, a challenging scenario where different triggers lead to misclassification in multiple targeted categories. The proposed method, called EUMC, replaces adaptive trigger generators with a pool of category-aware subgraph triggers sampled from the graph itself. Trigger assignment is based on attachment probability shifts to determine which subgraph is most effective for each category. The approach also introduces a "select then attach" strategy to maximize the stealthiness of triggers by optimizing trigger-node similarity. The method is empirically validated across six real-world node classification datasets and several defense strategies, showing improved attack success with negligible impact on clean node accuracy in comparison with existing single-category and multi-category backdoor attacks.

**Questions:**

1. Could the authors provide insight into the limits of the pool-based approach—what happens as the number of categories or trigger pool size is dramatically increased? Are there edge cases where the MC-STP approach fails to find effective triggers?

2. Is there any qualitative evidence or further statistical analysis showing how unnoticeable the constructed triggers are to a human or automated anomaly detector?

**Ethical Concerns:**

["NO or VERY MINOR ethics concerns only"]

**Limitations:**

See weaknesses

**Quality:**

3

**Strengths And Weaknesses:**

## Strengths

1. The paper addresses a relevant and challenging problem: graph backdoor attacks in the multi-category setting, which is not well studied in existing literature. The methodology leverages category-aware priors through a multi-category subgraph trigger pool, offering a more scalable approach than multiple independent trigger generators for each category. This is well-motivated both theoretically and, in Figure 1 and accompanying discussion, empirically justified.

2. Technical exposition is mostly clear, with well-defined notation and logical descriptions that make the approach accessible for replication by experts. Algorithm 1 includes detailed procedural steps.

3. Extensive experiments (Tables 1, 2, 3) across a variety of datasets (Cora, Pubmed, Bitcoin, Facebook, Flickr, OGB-arxiv) and defense scenarios provide compelling evidence for the claim that EUMC achieves high attack success rates (ASR) while maintaining clean node accuracy.

## Weaknesses

1. While the main logic and methodology are explained, certain practical implementation details are missing or under-explained in the core text, e.g., the precise role/impact of trigger size/threshold selection or prompt pool size. Figure 4 is only referenced at the end, with insufficient analytic commentary in the main text. This diminishes reproducibility and interpretability for non-domain experts.

2. The core idea of using subgraph triggers is referenced in prior work (e.g., SBA [29], Sheng et al. [32]), albeit for different settings (graph classification or single-category). The paper could do more to clarify and contrast what is uniquely enabled by the multi-category focus, both theoretically and in practice. There is room for more explicit differentiation from similar ideas in the literature.

3. The work considers several defenses, but does not include more recent or sophisticated graph backdoor detection approaches. Furthermore, there is little qualitative discussion or visualization of attack artifacts post-defense. How detectable are EUMC triggers via structural or statistical anomaly detection?

---

> ### Author Rebuttal · Authors · 2025-07-27
>
> We appreciate the reviewer‘s recognition of our problem's relevance, method scalability, clear technical exposition, and compelling experiments and thank the reviewer for the constructive comments. Regarding the concerns of the reviewer WCrP, we provide the following responses. We hope our responses address the concerns raised, and we welcome active discussions with reviewers.
>
> > **W1. More details about the precise role/impact of trigger size/threshold selection or prompt pool size.**
>
> We appreciate the reviewer's constructive comments. We have added a detailed explanation, which we briefly outline here:
>
> * **Trigger size**: Refers to the number of nodes in the trigger subgraph. A moderate number of nodes can effectively contain class structural information and class node information. If the size is too small, it fails to convey useful attack-related information; if too large, it may disrupt the degree distribution of the original graph structure, making it more likely to be pruned by defense strategies. Both scenarios can lead to a decrease in ASR.
>
> * **Threshold selection**: We select the most suitable trigger from the trigger pool for attacking the target node, and only connect nodes with similarity above the threshold to the target node, thereby enhancing the stealth of the attack. For instance, in arxiv dataset, only nodes with a similarity above 0.8 can connect to the target node, which significantly improves the stealth of the attack.
>
> * **Prompt pool size**: We apologize for the typo; this refers to "trigger pool size"—the number of triggers per class. For example, in 3-class Pubmed, we use 20 triggers per class, structured as {label1: [t1...t20], label2: [t1...t20], label3: [t1...t20]}.   A proper size enables effective multi-class learning with diverse structures. Too small: insufficient class learning and monotonous subgraphs. Too large: overlapping information and intra-class subgraph conflicts. Both reduce ASR and increase vulnerability to defense pruning.
>
> > **W2. Contribution and uniqueness about our method**
>
> * **Multi-category setting**: To our knowledge, this work is the first to systematically address **multi-category graph backdoor attacks** in GNN-based node classification—a critical yet underexplored scenario. Unlike existing graph backdoor methods focused solely on single-category misclassification, we target the more challenging task of inducing misclassification for multiple categories using distinct triggers without cross-interference. Experiments validate our method’s effectiveness, validity, and stealth.
> * **Trigger pool with rich prior knowledge**:
>   * **Diverse and authentic structures**: Triggers are sampled directly as subgraphs from the original graph, **preserving its intrinsic structural patterns and feature distributions**. This avoids the "artificiality" of synthetic triggers, which often introduce unnatural topological/feature anomalies.
>   * **Explicit category awareness**: By sampling subgraphs from **a small subset (less than 5%)** of origin nodes, we embed **category-specific structural signatures** (by Eq.(1) ) into triggers. This inherently aligns triggers with their target categories’ topological traits, solving multi-category attacks’ core challenge: differentiating triggers across classes.
>   * **Scalability**: Unlike methods requiring separate trigger generators per category (computationally redundant as categories grow), our pool-based design scales efficiently. It dynamically supports multiple categories without re-training generators, **adapting to large-scale multi-class scenarios**.
> * **Stealthiness** is prioritized via a **"select then attach" strategy**, outperforming existing methods in preserving the graph’s natural properties:
>   * Unlike UGBA and DPGBA (which disrupt the original degree distribution by forcing single-edge trigger-host connections), our threshold-based selection only attaches triggers to host nodes with high structural/feature similarity. By retaining the original graph’s subgraph structures and aligning trigger-host similarity, triggers blend **more naturally into the topology**. This reduces detection risk from both structural anomaly detectors (e.g., monitoring topological outliers) and feature-based defenses (e.g., flagging unnatural feature mismatches).
>
> > **W3 & Q2. Further statistical analysis showing how unnoticeable the constructed triggers are to a human or automated anomaly detector? The work considers several defenses, but does not include more recent or sophisticated graph backdoor detection approaches. Furthermore, there is little qualitative discussion or visualization of attack artifacts post-defense. How detectable are EUMC triggers via structural or statistical anomaly detection?**
>
> * Among the defense strategies we tested, the OD strategy is a common automated anomaly detector. Specifically, the OOD detector (OD), trained on the poisoned graph, can identify triggers through reconstruction loss: by removing nodes with high reconstruction losses, the feature distribution of nodes can remain within the normal domain, preserving the data representation integrity. **As shown in Tab. 2**, ASR does not decrease significantly under OD detection, indicating that the defense strategy does not prune injected triggers excessively.
>
> * Futhermore, **in Fig. 3**, we have also analyzed the similarity distribution between the injected triggers and the original graph on the Arxiv dataset. It can be observed that the trigger similarity of our method not only falls within a reasonable range but also exhibits a distribution consistent with that of the original graph. This demonstrates that our method **can effectively evade defense strategies such as feature anomaly pruning and feature statistical distribution detectors**, while **not disrupting the similarity distribution of the original graph**.
>
> * We have also statistically analyzed the pruning ratio of OD, Prune, and Prune+LD across multiple datasets in **Table R1**. We can observe that existing defense methods can prune only a small portion of trigger nodes (0%-5%) but are generally unable to completely prune the entire trigger, which illustrates the stealthiness of our method. Furthermore, in **Table R2**, we tested our method on several recent detector approaches, further demonstrating the unnoticeable nature of our triggers.
>
> **Table R1**: The effectiveness of our method post-defense. 2.7%(0.2%) represents (node prune\|trigger prune)
>
> | dataset   | Prune      | Prune+LD   | OD         |
> | --------- | ---------- | ---------- | ---------- |
> | Cora      | 2.7%(0.2%) | 2.3%(0.2%) | 1.7%(0.0%) |
> | Pubmed    | 1.5%(0.1%) | 1.7%(0.3%) | 1.5%(0.2%) |
> | Bitcoin   | 0.7%(0.0%) | 1.2%(0.1%) | 1.0%(0.1%) |
> | Facebook  | 4.3%(0.8%) | 3.9%(0.7%) | 2.7%(0.4%) |
> | Flickr    | 3.6%(0.4%) | 3.2%(0.4%) | 1.8%(0.2%) |
> | OGB-arxiv | 3.2%(0.6%) | 2.8%(0.3%) | 2.8%(0.4%) |
>
> **Table R2**: The ASR(Clean Acc) under different detector approaches.
>
> | Method    | Cora       | Pubmed     | Bitcoin    | Facebook   | Flickr     | OGB-arxiv  |
> | --------- | ---------- | ---------- | ---------- | ---------- | ---------- | ---------- |
> | CONAD [1] | 95.3(81.7) | 94.9(83.9) | 89.1(78.3) | 91.3(83.6) | 90.2(44.7) | 82.8(65.2) |
> | GAAN [2]  | 96.1(82.1) | 95.5(84.2) | 88.3(78.3) | 90.9(83.3) | 90.3(45.2) | 83.1(65.1) |
> | XGBD [3]  | 95.7(81.4) | 95.3(84.1) | 88.6(78.3) | 91.7(83.6) | 89.9(44.8) | 83.0(65.1) |
>
> Reference:
>
> [1] Chen, Henxing, et al. "Generative adversarial attributed network anomaly detection". Conference on Information and Knowledge Management (2020).
>
> [2] Xu, Zhiming, et al. "Contrastive attributed network anomaly detection with data augmentation". Pacific-Asia Conference on Knowledge Discovery and Data Mining (2022).
>
> [3] Guan, Zihan, et al. "XGBD:Explanation-Guided Graph Backdoor Detection". arXiv preprint (2023).
>
> > **Q1. Could the authors provide insight into the limits of the pool-based approach—what happens as the number of categories or trigger pool size is dramatically increased? Are there edge cases where the MC-STP approach fails to find effective triggers?**
>
> * Unlike the single-structure and feature-dependent trigger generators in SBA, UGBA, and DPGBA, our pool-based method can effectively provide triggers with diverse structures, features, and class-specific knowledge as the pool size increases, addressing the issue where triggers struggle to attack different classes. However, as shown **in Tab. 2**, **with the increase in the number of categories**, the decrease in GNN clean accuracy leads to **insufficient learning of class information** by triggers, thereby reducing ASR. Additionally, **in Fig. 4a**, as the pool size increases to a certain extent, ASR stabilizes. This may be because triggers designed for the same class may inadvertently **learn overlapping information** due to inherent variability among nodes in that class, increasing the difficulty of learning highly specific and effective triggers. **With an increase in the size of the trigger pool**, the attack effectiveness and stealthiness are initially improved. However, if the pool becomes excessively large, it will lead to **increased subsequent computational overhead**, such as in the "select then attach" strategy. Additionally, it may contain **redundant subgraphs** that are isomorphic or near-isomorphic.
> * In **Fig. 3**, we show the similarity distribution between injected triggers and the original graph on the Arxiv dataset. Compared with methods like UGBA and GTA, our approach can effectively constrain trigger similarity within the range of original graph similarity and align with its distribution, demonstrating the stealthiness of our method. Meanwhile, in **Table R1**, we present the pruning ratios of triggers under different defense strategies—only a minimal proportion (**0.5%**) of triggers and (**4%**) of nodes are pruned, indicating that the MC-STP approach rarely fails to find effective triggers.

---

> ### Author Response · Authors · 2025-08-07
>
> Dear reviewer:
>
> We sincerely appreciate your thoughtful review. As the author-reviewer discussion period is drawing to a close, we would be truly grateful to receive any further feedback you might have. If there are still any concerns that remain unaddressed, we would be more than happy to continue the discussion.

---

> > ### Comment · Reviewer_WCrP · 2025-08-07
> >
> > Thanks for the clarification, and my concerns are mostly addressed. I decide to maintain my score and suggest acceptance.

---

> > > ### Author Response · Authors · 2025-08-07
> > >
> > > Dear Reviewer WCrP,
> > >
> > > Thank you sincerely for taking the time to carefully read our rebuttal. Your thorough consideration means a great deal to us. We are particularly encouraged by your recognition of our work, as it validates the efforts we’ve invested in refining our research.
> > >
> > > Best regards,
> > >
> > > The Authors of Submission 16516

---

### Official Review · Reviewer_yDtx · 2025-07-03

**Clarity:** 3
**Significance:** 2
**Originality:** 3
**Rating:** 5
**Confidence:** 3

**Summary:**

This paper proposes a novel multi-class graph backdoor attack method by constructing a multi-class subgraph trigger pool and adopting a "select then attach" strategy to connect appropriate triggers to the target nodes, ensuring these modifications remain inconspicuous. Experimental results demonstrate high attack success rates across various classification scenarios and showcase the effectiveness and stealthiness of the proposed method on multiple real-world datasets. Additionally, the paper discusses potential future research directions, such as exploring scalable trigger generation strategies and defense mechanisms.

**Questions:**

1. Detailed Steps for Trigger Selection Algorithm: It is expected that the authors provide more detailed explanations of the specific algorithm steps for selecting particular triggers. Especially in multi-class attack scenarios, how to ensure the effectiveness and stealthiness of the triggers?
2. Effectiveness of Defense Strategies in Practical Applications: In practical application scenarios, how effective are the proposed defense strategies (e.g., Prune, Prune+LD, and OD)? Are there plans to test these strategies on larger or more complex datasets?

**Ethical Concerns:**

["NO or VERY MINOR ethics concerns only"]

**Final Justification:**

Thanks for the authors' detailed clarification, which has effectively addressed the key points of concern raised earlier. I have increased my rating.

**Limitations:**

1. Some experimental details (e.g., specific algorithm steps, dependency configurations) may require more explanation to allow other researchers to fully reproduce the results.
2. Some key technical details (e.g., how specific triggers are selected in the algorithm steps) may need more elaboration and examples to help readers better understand the implementation process.

**Paper Formatting Concerns:**

No major formatting issues were found in the paper.

**Quality:**

3

**Strengths And Weaknesses:**

Strengths:
Quality: The study designs an innovative multi-class graph backdoor attack method and validates its effectiveness through extensive experiments.
Clarity: The paper is well-structured with detailed method descriptions that are easy to understand and reproduce.
Significance: Proposes new attack methods, providing an important empirical foundation for security research in Graph Neural Networks (GNNs).
Originality: This is the first attempt to construct a trigger pool suitable for multiple target classes, showcasing strong innovation.
Weaknesses:
1. Insufficient experimental details: Some experimental details (e.g., specific algorithm steps, dependency configurations) may require more explanation to allow other researchers to fully reproduce the results.
2. Inadequate explanation of technical details: Although the overall structure is clear, some key technical details (e.g., how specific triggers are selected in the algorithm steps) may need more elaboration and examples to help readers better understand the implementation process.
3. Although the experimental results are convincing, the impact in practical application scenarios requires further exploration.
4. Some of the defense strategies adopted in this study draw on pre-existing techniques, which indicates a lack of thorough independent innovation.

---

> ### Author Rebuttal · Authors · 2025-07-27
>
> We appreciate the reviewer's recognition of our novel method, its effectiveness shown in experiments, the paper's clarity, significance for GNN security, and strong originality and we thank the reviewer for the constructive comments. Regarding the concerns of the reviewer yDtx, we provide the following responses. We hope our responses address the concerns raised, and we welcome active discussions with reviewers.
>
> > **W1 & W2 & Q1. Details about algorithm steps, dependency configurations and how to ensure the effectiveness and stealthiness of the triggers.**
>
> We appreciate the reviewer for the constructive comment. We have detailed our experiments and algorithm in the revised version. We provide the discussion as follows:
>
> * **Dependency configurations**: We have specified the hyperparameter settings in **Section 5.1 Implementation Details**. Additionally, we will provide more detailed parameter explanations and experimental environment configurations in the open-source code repository.
> * **Algorithm steps**: We further elaborate on the steps as follows:
>   *  **Construction of MC-STP**:
>      1.  As mentioned in **Section 4.1**, we initially randomly select several nodes from the unlabeled graph, $V_U$, to serve as central nodes. These central nodes account for a small proportion of the original graph, and we do not sample all possible subgraphs. Instead, we set thresholds for the total number of sampled subgraphs and the number of subgraphs sampled per central node.
>      2.  All obtained subgraphs are clustered to generate  $V_P$ subgraphs, which are used as the initialization of MC-STP.
>      3.  We assign target attack labels to each trigger based on how their effect on the prediction of attached nodes. In detail, we use APS to determine the target attack label of the trigger. (a) A classifier is used to predict the probability distributions of node labels **before and after trigger injection**, and the difference between these distributions is calculated. (b) By summing and averaging the label probability shifts across multiple nodes, we obtain **an APS vector**, which **represents the extent of the trigger’s impact on all classes**. (c) Finally, we obtain MC-STP structured as {label1: [t1, t2...], label2: [t1, t2...], ...}.
>      4.  **For example**, using the OGB-arxiv dataset (with 169,343 nodes and 1,166,243 edges), we first select 4000 central nodes, and each central node sampled **10 subgraphs**, resulting in a total of **40,000 sampled subgraphs**. After **k-means clustering**, we obtained an MC-STP containing 800 triggers. We then assigned **target attack labels** to the 800 triggers using Eq. (1) in Section 4.1 based on their Attachment Probability Shift(**APS**).
>          A trigger $t$ is attached to $N$ nodes. A GNN classifier then calculates the probability distributions $p_{bef}^{t, i}$ and $p_{aft}^{t, i}$ before and after the trigger $t$ is injected into the nodes. Here, $p_{bef}^{t, i}, p_{aft}^{t, i} \in \mathbb{R}^{N_{cls}}$ represents the predicted probabilities for each class. At this point, we compute $\text{APS}^{t} = \frac{1}{N} \sum_{i=1}^{N}|p_{bef}^{t, i} - p_{aft}^{t, i}|$ to obtain the $\text{APS}^{t} \in \mathbb{R}^{N_{cls}}$. The $\text{APS}^{t}$ indicates the impact of the trigger $t$ on each class, and we select the category with the greatest influence as the **target attack label** $y_t=\arg\max(\text{APS}^{t})$ for trigger $t$, indicating that $t$ has the most significant impact on category $y_t $ and can be more stealthily disguised as nodes of category $y_t $.
>   *  **Trigger selection and attachment**:
>      *   In Section 4.2, we detailed the trigger selection process. For an attacked node $ v_i \in V_P $, we first calculate the similarity between $v_i $ and all candidate triggers of the target class using Eq. (2), then select the trigger with the highest similarity for injection.
>      *   **For example**, given 20 candidate 5-node triggers, we compute the **cosine similarity** between the attacked node $v_i $ and each of the 5 nodes in each trigger, sum these similarities to generate a score in Eq. (2), and finally select the trigger with the highest score for injection. After choosing the suitable trigger $t$ for injection, we will continue to consider the edges connected between the attacked node $v_i$ and nodes of trigger $t$. To ensure attack effectiveness and stealthiness, we only connect nodes within the trigger $t$ to the attacked node $v_i $ if their **cosine similarity exceeds a predefined threshold** $\tau_a$.
>   *  **Optimization**: To **guarantee the effectiveness and stealthiness of the triggers**, we adopt **a bi-level optimization** approach (Eq. (3), (4), and (5) in Section 4.3). **The inner loop** optimizes the accuracy of the surrogate model on both clean nodes and attacked nodes via Eq. (3). **The outer loop** ensures attack effectiveness and stealthiness, constrained by the attack effectiveness objective (Eq. (4)) and stealthiness objective (Eq. (5)). Our method achieves state-of-the-art results across multiple GNN models and defense strategies, demonstrating the effectiveness and stealthiness of MC-STP in multi-class attack scenarios.
>
> > **W3. The impact in practical application scenarios requires further exploration.**
>
> We appreciate the reviewer for the constructive comment. We have detailed our **case study** about practical application scenarios in the revised version. We provide the discussion as follows:
>
> **Scenario:** Node classification on Facebook identifies account types as $C_1$ (politicians), $C_2$ (governmental organizations), $C_3$ (television shows), and $C_4$ (companies), enabling differentiated governance (e.g., preferential content distribution for politicians, restricted promotions for non-companies).
>
> * **Attack Goals**: Attackers aim for **multi-category precise tampering**:
>   * Disguise uncertified $C_4$ (companies) as $C_1$ (politicians) to bypass promotion restrictions and gain exposure.
>   * Disguise low-engagement $C_3$ (TV shows) as $C_2$ (governmental organizations) to exploit public trust for misleading content.
>   * Avoid detection by platform defenses (e.g., topology anomaly checks, certification validation).
> * **Practical Role of MC-STP**:
>   * Traditional single-category methods (e.g., GTA) fail to target $C_1$ and $C_2$ simultaneously, with over-generic triggers easily detected.
>   * MC-STP samples category-adaptive triggers: mimicking politician accounts’ dense public interaction topology for $C_1$, and governmental organizations’ official association structures for $C_2$.
>   * The "select-then-attach" strategy ensures triggers align with targets’ original edge structures (e.g., integrating politician-style interactions into company node), meeting real-world stealth needs.
>
> > **W4 &Q2. Some of the defense strategies adopted in this study draw on pre-existing techniques, which indicates a lack of thorough independent innovation. In practical application scenarios, how effective are the proposed defense strategies (e.g., Prune, Prune+LD, and OD)? Are there plans to test these strategies on larger or more complex datasets?**
>
> * **Our study focuses on achieving stealthy and efficient backdoor attacks in multi-category scenarios.** The use of pre-existing popular defense strategies is solely to demonstrate the effectiveness and stealth of our attack. While defense research is currently outside the scope of our study, **we will consider exploring ways to further enhance defense mechanisms in the future.**
>
> * Following [1] and [2], we adopt popular graph domain defense strategies such as Prune, Prune+LD, and OD, which maintain the cleanliness of graph data by pruning graph features and structures. To verify the effectiveness of our method, we statistically analyzed the pruning ratio of nodes in triggers and triggers by these defense strategies, demonstrating the effectiveness and stealth of our approach.
>
> * Among the tested datasets, OGB-arxiv is a large citation network containing 169,343 nodes, 1,166,243 edges, and 40 categories. Results in **Table R1** also show that our model effectively evades defenses in 40-class attacks. **We appreciate your suggestion and will further test our method on other more larger and complex datasets in future work.**
>
> **Table R1**: The effectiveness of our method post-defense. 2.7%(0.2%) represents (node prune|trigger prune)
>
> * | dataset   | Prune      | Prune+LD   | OD         |
>   | --------- | ---------- | ---------- | ---------- |
>   | Cora      | 2.7%(0.2%) | 2.3%(0.2%) | 1.7%(0.0%) |
>   | Pubmed    | 1.5%(0.1%) | 1.7%(0.3%) | 1.5%(0.2%) |
>   | Bitcoin   | 0.7%(0.0%) | 1.2%(0.1%) | 1.0%(0.1%) |
>   | Facebook  | 4.3%(0.8%) | 3.9%(0.7%) | 2.7%(0.4%) |
>   | Flickr    | 3.6%(0.4%) | 3.2%(0.4%) | 1.8%(0.2%) |
>   | OGB-arxiv | 3.2%(0.6%) | 2.8%(0.3%) | 2.8%(0.4%) |
>
> Reference:
>
> [1] Zhang, Zhiwei, et al. "Rethinking Graph Backdoor Attacks: A Distribution-Preserving Perspective" Proceedings of the 30th ACM SIGKDD Conference on Knowledge Discovery and Data Mining (2024).
>
> [2] Ding, Yuanhao, et al. "SPEAR: A Structure-Preserving Manipulation Method for Graph Backdoor Attacks" Proceedings of the ACM Web Conference (2025).

---

> ### Author Response · Authors · 2025-08-07
>
> Dear reviewer:
>
> We sincerely appreciate your thoughtful review. As the author-reviewer discussion period is drawing to a close, we would be truly grateful to receive any further feedback you might have. If there are still any concerns that remain unaddressed, we would be more than happy to continue the discussion.

---

> ### Author Response · Authors · 2025-08-07
>
> Dear Reviewer yDtx,
>
> Thank you very much for taking the time to carefully read our rebuttal. We truly appreciate the attention and thought you have dedicated to our responses, as your insights are invaluable to helping us improve our work.
>
> Please feel free to reach out if there are any further questions, concerns, or points you would like to discuss. We are more than happy to provide additional clarification or engage in further dialogue to address any remaining issues.
>
> Once again, thank you for your continued guidance and support.
>
> Best regards,
>
> The Authors of Submission 16516

---

### Official Review · Reviewer_StDL · 2025-07-03

**Clarity:** 2
**Significance:** 3
**Originality:** 2
**Rating:** 4
**Confidence:** 2

**Summary:**

This paper addresses the challenge of multi-category graph backdoor attacks on node classification tasks, where existing methods (e.g., GTA, UGBA) fail to scale effectively as the number of target categories increases. The authors propose EUMC, a novel framework that leverages subgraph triggers sampled from the original graph to construct a Multi-Category Subgraph Triggers Pool (MC-STP). Experiments across six datasets (Cora, Pubmed, Bitcoin, Facebook, Flickr, OGB-arxiv) and three GNN architectures (GCN, GAT, GraphSAGE) show EUMC outperforms baselines in Attack Success Rate (ASR) under defenses (Prune, Prune+LD, OD), particularly for large category sets.

**Questions:**

- Experiments fix trigger size at 5 nodes (Sec 5.1). Why is this optimal? Does performance plateau for larger triggers (Fig 4b) due to detectability or GNN aggregation limits?
- Sampling subgraphs via BFS (Sec 4.1) may not scale to billion-edge graphs. Can MC-STP initialization be accelerated?
- Why does ASR decrease as categories grow (Table 1)? Is this due to inter-category trigger interference or reduced subgraph uniqueness?

**Ethical Concerns:**

["NO or VERY MINOR ethics concerns only"]

**Final Justification:**

Thank the authors for their responses. My questions are largely resolved, and I will keep my rating as borderline acceptance.

**Limitations:**

Yes.

**Paper Formatting Concerns:**

No.

**Quality:**

2

**Strengths And Weaknesses:**

Strengths:
- The manuscript is well-structured with clear figures (e.g., Fig 1, 2) and Algorithm 1. Appendix details (e.g., time complexity, hyperparameters) aid reproducibility.
- Solves a critical gap in multi-category backdoor attacks, enabling precise control over diverse target classes—a need for real-world adversarial scenarios.

Weaknesses:
- Limited exploration of transferability to non-surrogate models. Experiments on robust GNNs (GNNGuard, RobustGCN) lack comparison to baselines.
- ASR still declines with increasing categories (e.g., Table 1, Flickr), though less severely than baselines. No theoretical analysis of this trend.

---

> ### Author Rebuttal · Authors · 2025-07-27
>
> We appreciate the reviewer's recognition of our framework addressing the multi-category graph backdoor attack challenge, its effectiveness shown in experiments, the well-structured manuscript, and its role in filling a critical gap and we thank the reviewer for the constructive comments. Regarding the concerns of the reviewer StDL, we provide the following responses. We hope our responses address the concerns raised, and we welcome active discussions with reviewers.
>
> > **W1. Limited exploration of transferability to non-surrogate models. Experiments on robust GNNs (GNNGuard, RobustGCN) lack comparison to baselines.**
>
> * In our experiments, only the GCN is exploited as the surrogate model to optimize our multi-category trigger pool. Therefore, the experimental results on GAT, GraphSage, GraphTransformer, GNNGuard and RobustGCN can be regarded as the exploration of transferability.
>   On these above non-surrogate models, our model shows great performance in terms of attack success rate(ASR).
> * **In Appendix E, Tab. 8 and 9**, we compare the performance of all baselines on GNNGuard and RobustGCN. Compared to DPGBA, our method outperforms by **12.3%** and **3.4%** on GNNGuard and RobustGCN respectively. Compared to UGBA, the improvements are **26.7%** and **13.9%** respectively, demonstrating our method's stealthiness and effectiveness in multi-class attack scenarios.
>
> > **W2 & Q3. ASR still declines with increasing categories (e.g., Table 1, Flickr), though less severely than baselines. No theoretical analysis of this trend. Why does ASR decrease as categories grow (Table 1)? Is this due to inter-category trigger interference or reduced subgraph uniqueness?**
>
> We appreciate the reviewer's constructive comment. The reasons that lead the ASR still declines with increasing categories can be summarized as:
>
> * **Classifier performance limitations**: As shown in **Tab. 2**, the clean accuracy of GNN classifiers decreases significantly with the increase in the number of categories. Compared to the 3-class Pubmed dataset with an accuracy of **84%**, the multi-class Flickr and Arxiv datasets only achieve **44%** and **65%** accuracy, respectively. Since GNNs themselves struggle to distinguish over half of the nodes on Flickr, **the class-specific information learned by MC-STP is limited**, which in turn leads to a decline in attack accuracy.
> * **Differences in dataset characteristics**:  For multi-class graph datasets, each class exhibits distinct structural and information patterns, and even nodes within the same class may have variations. This leads to **potential conflicts in learning class-specific triggers for the same class**. Additionally, triggers designed for the same class may inadvertently **learn overlapping information** due to the inherent variability among nodes in that class, increasing the difficulty of learning highly specific and effective triggers.
>
> > **Q1. Experiments fix trigger size at 5 nodes. Why is this optimal? Does performance plateau for larger triggers due to detectability or GNN aggregation limits?**
>
> * **In Appendix Fig. 4b**, we conduct an ablation study on the number of trigger nodes. The performance improves as the number of nodes increases, but it plateaus when exceeding 5 nodes while maintaining overall strong performance. Thus, we uniformly set the trigger size to 5 nodes.
> * We attribute the performance plateau after 5 nodes to **two reasons**:
>   - First, it is limited by **graph information propagation**. If the trigger is too large, the GNN’s information propagation cannot cover all node information, resulting in the waste of some node information.
>   - Second, it is related to **the degree distribution of the original data**. As observed **in Table R1**, the degree distribution of most datasets is mostly below 7. Thus, the maximum degree of a node in a five-node trigger is 5 under fully-connection setting (4 within the triiger plus 1 for attacked node). Moreover, when the number of nodes in the trigger is too large, some nodes within the trigger may not conform to the degree distribution of the original graph, which may **increase detectability** and result in **being pruned**.
>
> **Table R1**: The statistics on the degree of different graph dataset.
>
> | dataset   | nodes   | edge      | degree |
> | --------- | ------- | --------- | ------ |
> | Cora      | 2,708   | 5,429     | 3.9    |
> | Pubmed    | 19,717  | 44,338    | 4.5    |
> | Flickr    | 89,250  | 899,756   | 10.1   |
> | OGB-arxiv | 169,343 | 1,166,243 | 6.9    |
>
> > **Q2. Sampling subgraphs via BFS may not scale to billion-edge graphs. Can MC-STP initialization be accelerated?**
>
> We thank the reviewer for the suggestion to discuss the efficiency of MC-STP. We provide the following discussion:
>
> * First, subgraph sampling via BFS is a **pre-processing step that only needs to be performed once**. In subsequent processes (trigger selection, trigger optimization and attacking), we can reuse the previously sampled results without re-conducting BFS.
>
> * Second, as mentioned in **Section 4.1**, we initially randomly select several nodes from the unlabeled graph,$V_U$, to serve as central nodes. These central nodes account for **a small proportion** of the original graph, and we do not sample all possible subgraphs. Instead, we set thresholds for the total number of sampled subgraphs and the number of subgraphs sampled per central node. **This ensures high efficiency even on billion-edge graphs**.
>
> * Third, we analyzed **the time complexity of the MC-STP Initialization**. Let $h$ denote the embedding dimension, $n$ represent the number of nodes in each trigger, $K$ be the number of target categories, and $|\mathcal{N}_S|$ be the number of subgraphs to extract. The cost of constructing the MC-STP is approximately $O(nh|\mathcal{N}_S| + Kh|\mathcal{N}_S| + Kh)$, which covers extracting $|\mathcal{N}_S|$ subgraphs with $n$ nodes each and clustering them into $K$ clusters. Since subgraph and feature extraction is the dominant cost, the overall time complexity is approximately $O(Kh|\mathcal{N}_S|)$. Therefore, the complexity of the MC-STP initialization is proportional to the number of sampled subgraphs. **Furthermore, subsequent experiments and analyses demonstrate that MC-STP initialization is not time-consuming.**
>
> * For example, the OGB-arxiv dataset has 169,343 nodes and 1,166,243 edges. **In the pre-processing step,** we selected 4000 central nodes, and each central node was used to sample 10 subgraphs, resulting in a total of **40,000 sampled subgraphs**. The number of nodes used accounts for **2.3%** in the original graph. After clustering, we obtained MC-STP containing 800 triggers, so APS only needs to be calculated within these 800 triggers. **In the training step**, we only used **2.7% nodes with labels**. These proportions are very small, so they do not incur significant computational costs.
>
> * We have also conducted **efficiency tests on multiple datasets** in **Table R2**, demonstrating that subgraph sampling does not incur significant computational costs.
>
> **Table R2**: The time cost of construction MC-STP.
>
> | dataset   | sample central nodes | sample subgraph | time cost(s) |
> | --------- | -------------------- | --------------- | ------------ |
> | Cora      | 700                  | 7,000            | 78.6         |
> | Pubmed    | 1,200                 | 12,000           | 114.7        |
> | Bitcoin   | 1,500                 | 15,000           | 138.2        |
> | Facebook  | 1,600                 | 16,000           | 156.8        |
> | Flickr    | 2,100                 | 21,000           | 231.3        |
> | OGB-arxiv | 4,000                | 40,000          | 423.7        |

---

> ### Author Response · Authors · 2025-08-07
>
> Dear reviewer:
>
> We sincerely appreciate your thoughtful review. As the author-reviewer discussion period is drawing to a close, we would be truly grateful to receive any further feedback you might have. If there are still any concerns that remain unaddressed, we would be more than happy to continue the discussion.

---

> ### Comment · Reviewer_StDL · 2025-08-08
>
> Thank the authors for their responses. My questions are largely resolved, and I will keep my rating as borderline acceptance.

---

> > ### Author Response · Authors · 2025-08-08
> >
> > Dear Reviewer StDL,
> >
> > Thank you sincerely for taking the time to carefully read our rebuttal. Your thorough consideration means a great deal to us. We are particularly encouraged by your recognition of our work, as it validates the efforts we’ve invested in refining our research.
> >
> > Best regards,
> >
> > The Authors of Submission 16516

---

### Author Response · Authors · 2025-08-06

Dear Reviewers,

We hope this message finds you well. We are writing to gently follow up on our rebuttal submitted for our paper in response to your valuable feedback.

We greatly appreciated your insightful comments and concerns, which helped us refine our work further. In our rebuttal, we addressed each of your points in detail and hoped to clarify any ambiguities.

If you have had the chance to review our response, we would be grateful to know if there are any remaining questions or additional points you would like us to elaborate on. Please feel free to let us know—we are happy to provide further clarification to support your evaluation.

Thank you again for your time and effort in reviewing our work. We understand you are likely very busy, and we appreciate your consideration.

Best regards,

The Authors of Submission 16516

---

### Note · Authors · 2025-08-13

Dear AC:

Thank you for your time in evaluating our submission.
We have addressed the concerns of reviewer StDL, yDtx, WCrP and Q1/3 of ecxW. Regarding Q2 of ecxW on APS class alignment, we empirically demonstrate it. To illustrate APS class alignment, we provide the following proof.
>**Notation**
* $C$:Random variable for node class.
* $A$:Binary intervention variable for trigger.
* $G$:Computation graph of a node.
* $P_c(G,t,0/1)=P(C=c\mid G,A=0/1)$:Probability of class $c$ for $G$ without/with trigger.
* $H(C\mid G,A=0/1)$:Conditional entropy of the class distribution without/with attack.
* $I(C;A\mid G)$:Conditional mutual information between class $C$ and trigger intervention $A$, on $G$.

>**Goal**

$c_t=\arg\max_c(\text{APS}_t)_c$ maximizes the mutual information(MI) $I(C;A\mid G)$, i.e. APS provided class maximize the trigger-class mutual information.
>**Assumptions**

`A1-Calibration`:
$f_{\theta_c}(G)_c=P(C=c\mid G)$(Model outputs posterior probabilities.)

`A2-Trigger Effectiveness`:
Exists $c_t$ and  $\delta>0$: $\forall$G, trigger t satisfies $P(C=c_t \mid A=1, G)-P(C=c_t \mid A=0, G) \geq \delta$

`A3-Uniform Intervention`:
$A\in\{0,1\}$ with $P(A=1)=P(A=0)=0.5$

`A4-Node Expectation`:
APS = $\mathbb{E}_{V\sim\text{Uniform}(V_P)}[\cdot]$

>**Proof**

`1. MI Decomposition`

Definition: $I(C; A \mid G) = H(C \mid G) - H(C \mid A, G)$

Entropy: $H(C \mid A, G) = \sum_{a \in \{0,1\}} P(A = a)H(C \mid A = a, G)$

Apply A3 and $H(C\mid A=0,G)=H(C\mid G)$: $H(C\mid A,G)=0.5\cdot H(C\mid G)+0.5\cdot H(C\mid A=1,G)$

Simplify: $I(C;A\mid G)=H(C\mid G)-[0.5\cdot H(C\mid G)+0.5\cdot H(C\mid A=1,G)]=0.5[\underbrace{H(C\mid G)}_{\text{constant}}-H(C\mid A=1,G)]$

`2. MI Maximization`

For fixed $G$, $\max_{t}I(C;A\mid G)\Longleftrightarrow\min_{t}H(C\mid A=1, G)$ where $H(C\mid A=1,G)=-\sum_{c}P_{c}(G,t,1)\log P_{c}(G,t,1).$

`3. Connecting APS to MI`

Given$c_t=\arg\max_c\mathbb{E}_G[P_c(G,t,1)-P_c(G,t,0)],$ since $P_c(G,t,0)$ (noo attack) is fixed,
$\arg\max_c\mathbb{E}_G[P_c(G,t,1)-P_c(G,t,0)]=\arg\max_c\mathbb{E}_G[P_c(G,t,1)]$

Entropy minimization requires: $\min_t H(C\mid A=1,G)\iff\max_t(\max_c P_c(G,t,1)) \forall G$ Under $\sum_c P_c(G,t,1)=1$, maximizing $\mathbb{E}_G[P_c(G,t,1)]$ forces the distribution to peak, $\max_c\mathbb{E}_G[P_c(G,t,1)]\implies\min\mathbb{E}_G[H(C\mid A=1,G)]$.

Thus $\boxed{c_t=\arg\max_c(\text{APS}_t)_c\implies\max_tI(C;A\mid G)}$. So, **APS is empirical and theoretical class-aligned**.

Sincerely,

Authors of 16516

---

### Decision · Program_Chairs · 2025-09-17

**Decision:**

Accept (poster)

**Comment:**

Existing graph backdoor attack methods mainly focus on single attack categories and struggle to handle backdoor attacks across multiple categories as the number of target categories increases. This paper proposes a new approach for Effective and Unnoticeable Multi-Category (EUMC) graph backdoor attacks, leveraging subgraphs from the attacked graph as category-aware triggers to precisely control the target category.

Pros:
- The paper studies a less-studied problem of graph backdoor attacks in the multicategory setting. It proposes a new multi-class graph backdoor attack method
- The paper is well-structured and easy to follow
- Experimental results demonstrate the effectiveness of the proposed method

Cons:

Most of the reviewers’ concerns are addressed except the one below

- The Reviewer ecxW still has two major concerns after the rebuttal, i.e., the lack of mechanism-level evidence for class-specificity and the limited methodological novelty

As most of the reviewers agree on the novelty and contribution of the paper, the paper can be accepted. The authors should incorporate clarifications and extra experiments into the camera-ready version.